**Resource**

# Using CombiCells, a platform for titration and combinatorial display of cell surface ligands, to study T-cell antigen sensitivity modulation by accessory receptors

Ashna Patel [ID] [1], Violaine Andre[1], Sofia Bustamante Eguiguren[1], Michael I Barton [ID] [1], Jake Burton[1], Eleanor M Denham[1,3], Johannes Pettmann[1,4], Alexander M Mørch [ID] [2], Mikhail A Kutuzov [ID] [1], Jesús A Siller-Farfán[1], Michael L Dustin [ID] [2], P Anton van der Merwe [ID] [1] & Omer Dushek [ID] [1✉]

## Abstract

**Understanding cellular decisions due to receptor–ligand interactions at cell–cell interfaces has been hampered by the difficulty of independently varying the surface density of multiple different ligands. Here, we express the synthetic binder protein SpyCatcher, designed to form spontaneous covalent bonds with interactors carrying a Spytag, on the cell surface. Using this, we show that addition of different concentrations and combinations of native Spytag-fused ligands allows for the combinatorial display of ligands on cells within minutes. We use this combinatorial display of cell surface ligands—called CombiCells—to assess T cell antigen sensitivity and the impact of T cell co-stimulation and co-inhibition receptors. We find that the T cell receptor (TCR) displayed greater sensitivity to peptides on major-histocompatibility complexes (pMHC) than synthetic chimeric antigen receptor (CARs) and bi-specific T cell engager (BiTEs) display to their target antigen, CD19. While TCR sensitivity was greatly enhanced by CD2/CD58 interactions, CAR sensitivity was primarily but more modestly enhanced by LFA-1/ICAM-1 interactions. Lastly, we show that PD-1/PD-L1 engagement inhibited T cell activation triggered solely by TCR/pMHC interactions, as well as the amplified activation induced by CD2 and CD28 co-stimulation. The ability to easily produce cells with different concentrations and combinations of ligands should accelerate the study of receptor–ligand interactions at cell–cell interfaces.**

**Keywords** Receptor/Ligand Interactions; Cell–Cell Recognition; Surface Ligand Presentation; T Cell Antigen Sensitivity; Chimeric Antigen Receptors
**Subject Categories** Biotechnology & Synthetic Biology; Immunology; Methods & Resources

## Introduction

Direct cell–cell communication is a ubiquitous and essential process in multicellular organisms. It is critical during development and tissue maintenance, and underlies the proper functioning of the nervous and immune systems (Belardi et al, 2020). Communication at cellular interfaces relies on diverse families of surface receptors that transduce signals upon recognizing their ligands on the surface of other cells. When studying surface receptors that recognize ligands in solution (e.g., G-Protein Coupled Receptors, Receptor Tyrosine Kinases, and Cytokine Receptors), it is trivial to experimentally vary the concentration and combination of soluble ligands. In contrast, it is far more challenging to vary the concentration and combination of cell surface ligands. This technical limitation has hampered our ability to understand cell–cell recognition.

Arguably the most well studied form of cell–cell recognition is T cell antigen recognition. T cells continuously patrol and scan cells throughout the body, seeking abnormal antigens derived from pathogens and mutated proteins produced by cancer cells. T cell activation hinges on whether their T cell antigen receptors (TCRs) bind these antigens, usually in the form of peptides presented on major histocompatibility complexes (pMHCs). Crucially, the response of the T cell also depends on engagement of other "accessory" receptors which can enhance or inhibit the response (Chen and Flies, 2013; Dushek et al, 2012). Infected or cancerous cells can evade immune recognition by reducing the level of antigen they express on their cell surface (Siller-Farfan and Dushek, 2018). For example, relapses following chimeric antigen receptor (CAR)-T cell therapy are associated with decreases in levels of the target antigen CD19 on the surface of cancer cells (Majzner and Mackall, 2018). In addition, pathogen-infected and cancerous cells can evade immune recognition by changing the levels of ligands to accessory receptors (Abdul Razak et al, 2016; Larson et al, 2022; Wang et al, 2018). It follows that it is important to be able to investigate how

[1]Sir William Dunn School of Pathology, University of Oxford, Oxford OX1 3RE, UK. [2]The Kennedy Institute of Rheumatology, University of Oxford, Oxford OX3 7FY, UK. [3]Present address: EnaraBio Ltd, The Bellhouse Building, Oxford Science Park, Sanders Road, Oxford OX44GD, UK. [4]Present address: GlaxoSmithKline Pharmaceuticals, Rue de l'Institut 89, 1330 Rixensart, Belgium. ✉E-mail: omer.dushek@path.ox.ac.uk

T cell activation is regulated by the concentration of antigens and the combinations of accessory receptor ligands on the target cells.

The accessory receptors CD2, LFA-1, and CD28 are known to enhance T cell responses mediated by the TCR (Abu-Shah et al, 2020; Bachmann et al, 1999, 1997), but their contribution to T cell responses mediated by CARs remains less clear. This is challenging to study as it is difficult to manipulate the surface levels of CAR and accessory receptor ligands. Current methods rely on laborious genetic methods to produce cell lines with desired combinations/surface densities of the required ligands. However, the number of cell lines needed increases exponentially with the number of ligands and surface densities, if all combinations are to be tested, making such experiments impractical. Moreover, the method is susceptible to genetic drift between these cell lines that must be in culture for weeks or months, making it difficult to conclude with certainty that differences observed are actually the result of differences in ligand expression.

Here, we introduce a novel platform enabling the rapid production of cells expressing any combination and concentration of ligands, and we use it to study T cell activation via a native TCR, synthetic CARs, and bi-specific T cell engagers (BiTEs), and the contribution of accessory receptors.

# Results

## The purified extracellular domain of native ligands fused to Spytag can readily couple to cell surface Spycatcher

To enable the combinatorial display of ligands on cells (Combi-Cells), we reasoned that cell surface expression of the protein Spycatcher, which forms a spontaneous covalent bond with a peptide tag (Spytag) (Keeble and Howarth, 2020), could be used to couple the extracellular domain of purified ligands fused to Spytag (Fig. 1A). Consequently, we fused the C-terminus of Spycatcher to the extracellular hinge of human CD52 (hCD52; 7 aa), murine CD80 (mCD80; 20 aa), or a variant of mCD80 that contained fewer residues (mCD80-short; 6 aa). The rationale for coupling the C-terminus of Spycatcher to these short hinges is that it would be expected to maintain a compact conformation bringing Spytag fusion proteins close to the membrane. The CD52 and CD80 hinges are anchored to the cell surface through glycosylphosphatidylino-sitol (GPI) and a transmembrane domain, respectively. We transduced these surface Spycatchers into CHO-K1 cells and detected expression by coupling a purified fluorescent protein fused to Spytag (Spytag-mClover3, Fig. 1B). A titration of Spytag-mClover3 revealed that the hCD52 hinge surface Spycatcher expressed at the highest level and that saturation was achieved at approximately 1 µM of Spytag-mClover3. Importantly, we confirmed that all surface Spycatchers were mobile at the cell surface with diffusion coefficients typical for membrane proteins (Appendix Fig. S1). Given its higher expression, we used surface Spycatcher fused to the hinge of hCD52 for subsequent experiments.

T cell activation is known to be controlled in part by the accessory receptors CD2, LFA-1, and CD28, whose ligands are CD58, ICAM-1, and CD86 (or CD80), respectively. To study their individual contributions using surface Spycatcher, a target cell that does not express these ligands is required. Given that CHO-K1 cells are hamster ovary cells, they are not expected to express ligands that bind these receptors, with the exception of ICAM-1, which has

been shown to be functional (Milstein et al, 2008). Therefore, we used CRISPR to knockout hamster ICAM-1 before transducing surface hCD52-Spycatcher (Fig. 1C,D). We refer to these CHO-K1 ICAM-1$^-$ hCD52-Spycatcher$^+$ as CHO-K1 CombiCells.

We next designed constructs that contained the full extracellular domains of CD58, ICAM-1, CD80, and CD86 fused to a C-terminal Spytag (for coupling to Spycatcher) and Histag (for purification). We produced and purified these ligands and coupled them to CHO-K1 CombiCells before measuring their surface levels using flow cytometry (Fig. 1E). We found that each ligand can be coupled at levels ≳10-fold higher than those found on the T2 cell line, other cell lines, and primary T cells and macrophages (Fig. 1F). Indeed, the absolute number of ligands that can be coupled exceeded ~$10^6$ per cell (Fig. 1G). We found that coupled ligands had a cell surface lifetime of ≈7 h detected using ligand-specific or his-tag antibodies (Fig. 1H; Appendix Fig. S2).

## The accessory receptor CD2 primarily controls the sensitivity of a pMHC targeting TCR and CAR

To study the impact of accessory receptor ligands on T cell antigen sensitivity, we produced purified Spytag-pMHC by refolding HLA-A*02:01 fused to Spytag with $\beta_2$m and a peptide from the NY-ESO-1 cancer antigen (Fig. 2A). We generated primary human CD8 + T cells using a standard adoptive cell therapy protocol in which T cells are activated to proliferate with anti-CD3/CD28 beads and transduced with the NY-ESO1 specific 1G4 TCR (Pettmann et al, 2021). We performed a preliminary experiment by co-culturing T cells with CHO-K1 CombiCells loaded with different concentrations of Spytag-pMHC and each ligand. We first confirmed that the surface level of Spytag-pMHC can be varied without impacting the surface level of each Spytag-ligand (and vice versa). We confirmed this to be the case, provided that the total concentration of Spytag-proteins remained below 1 µM, which was the maximum concentration of Spytag-proteins subsequently used (Fig. 2B). We measured T cell activation by surface markers (4-1BB, CD69) and by secreted cytokines (IL-2, IFN-γ, and TNF-α) (Figs. 2C and EV1). We observed the expected increase in T cell activation with increasing concentrations of Spytag-pMHC and with increasing concentration of each Spytag-ligand, consistent with the co-stimulation function of LFA-1, CD2, and CD28. Therefore, this preliminary experiment confirmed that T cells can exploit Spytag-ligands to accessory receptors in recognizing Spytag-pMHC in a concentration-dependent manner.

We note that the impact of adding Spytag-ICAM-1 on T cell activation is largely absent on the parental CHO-K1 cell line prior to hamster ICAM-1 knockout (Figure EV2). This underlines the importance of removing endogenous ligands, and demonstrates that T cells can exploit endogenously expressed ligands when recognizing pMHC coupled to cell surfaces via Spycatcher/Spytag.

To directly compare the antigen sensitivity of the 1G4 TCR and a CAR, we used the D52N second-generation CAR (comprising CD28 hinge, transmembrane, and co-stimulation regions fused to the ζ-chain). This CAR recognizes the same NY-ESO-1 pMHC antigen (Maus et al, 2016) (Fig. 3A). Binding of pMHC tetramers suggests that it is expressed at slightly higher levels than the TCR (Fig. 3B). We stimulated these cells with CHO-K1 CombiCells presenting different concentrations of antigen, either alone, or in combination with a fixed concentration of one of the co-stimulation ligands. We measured

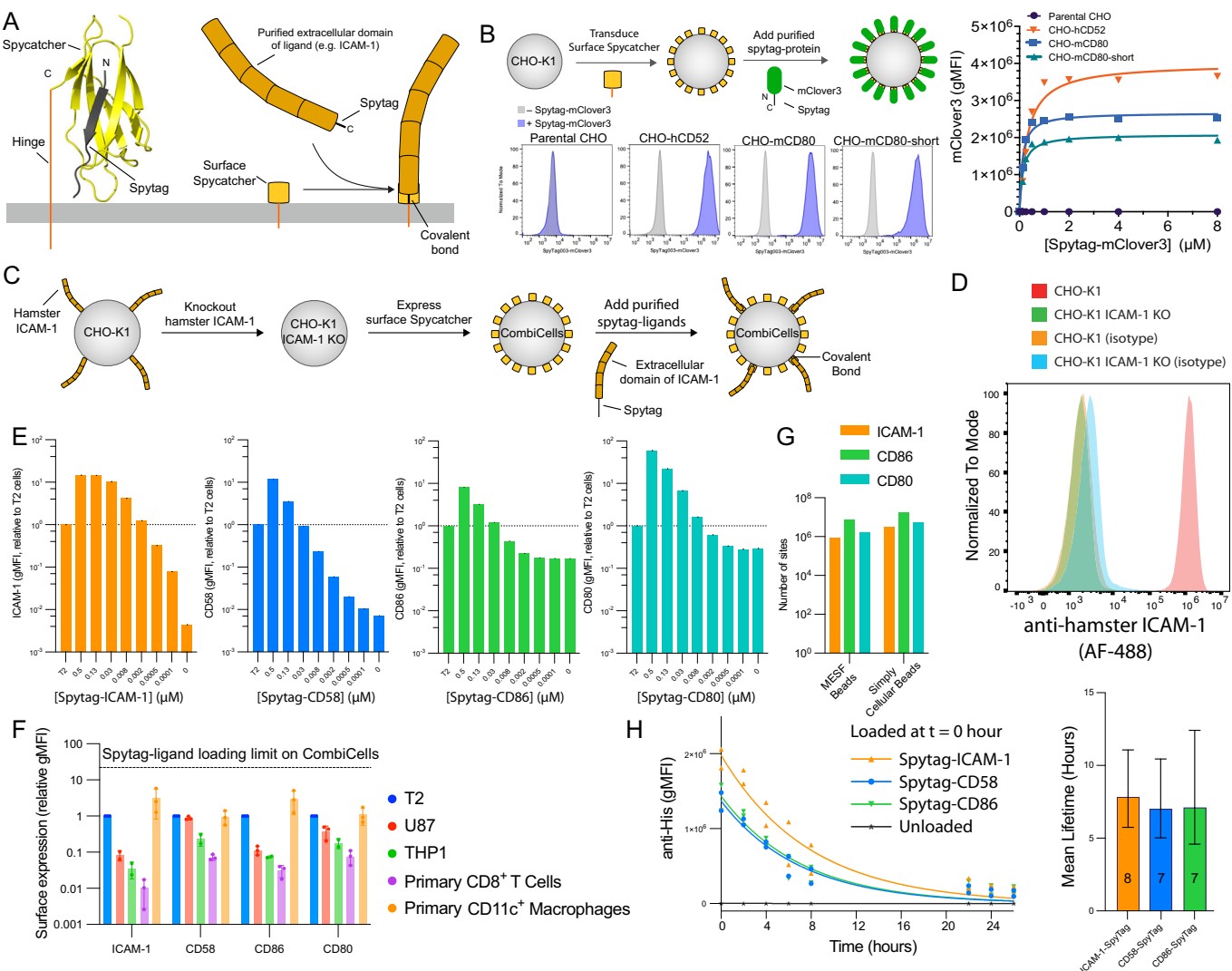

**Figure 1. Spycatcher can be expressed on the cell surface and can readily couple purified ligands to prominent T cell accessory receptors fused to Spytag.**

(A) Left: Structure of Spycatcher (yellow) coupled to Spytag (black) indicating the location of the hinge for surface display (C-terminus of Spycatcher) and the location of the extracellular domain of ligand (N-terminus of Spytag). Structure taken from PDB:4MLI. Right: Schematic of purified protein fused to Spytag coupling to surface Spycatcher. (B) CHO-K1 cells transduced with surface Spycatcher were coupled with purified mClover3 fused to Spytag and detected in flow cytometry. (C) Experimental workflow to generate CHO-K1 CombiCells. (D) Surface expression of hamster ICAM-1 on the indicated CHO-K1 cell line. (E) Expression of the indicated ligand on CHO-K1 CombiCells (coupled to Spycatcher) relative to native expression on T2 cells (horizontal line) detected by flow cytometry. (F) Expression of the indicated ligands on different cells relative to T2 cells ($N = 3$). (G) The absolute number of the indicated ligand per cell when coupled at 0.5 μM determined using the indicated calibration method. (H) CHO-K1 CombiCells were loaded with 0.1 μM of ligand and surface levels were measured over time (left, $N = 2$). An exponential fit is used to determine the mean lifetime (right). Data information: In (F), error bar are SD, and in (H), the error bars are standard error estimated from the exponential fit. Source data are available online for this figure.

surface markers (4-1BB, CD69), cytokines (IL-2, IFN-$\gamma$, and TNF-$\alpha$), and TCR/CAR downregulation (Figs. 3C–E and EV3).

In the case of the TCR, we found that all accessory receptors acted as co-stimulation molecules, but with different quantitative phenotypes. We found that CD2 substantially increased both antigen efficacy ($E_{max}$) and sensitivity ($P_{50}$) for all cytokines and some surface markers, and also increased TCR downregulation. LFA-1 had a more modest effect on antigen sensitivity for surface markers and TCR downregulation, but had no impact on cytokines. Finally, CD28 engagement increased antigen efficacy for IL-2 but had little other impact. These results show that, in expanded

human CD8$^+$ T cells, CD2 engagement has a larger impact over a broad range of responses than that of LFA-1 or CD28 engagement. In the case of the CAR, we found a similar qualitative pattern with CD2 imparting the largest co-stimulation effect. However, the quantitative impact was much more modest, with antigen sensitivity improving by 11 and 3.9-fold for 4-1BB and IL-2, respectively, compared to 230 and 46-fold for the TCR. As a result, the fold-difference in antigen sensitivity between the TCR and CAR increased from 30-fold when recognizing antigen alone to 300-fold or 120-fold when recognizing antigen in the presence of ligands for CD58 or LFA-1, respectively. The lack of any impact of extrinsic

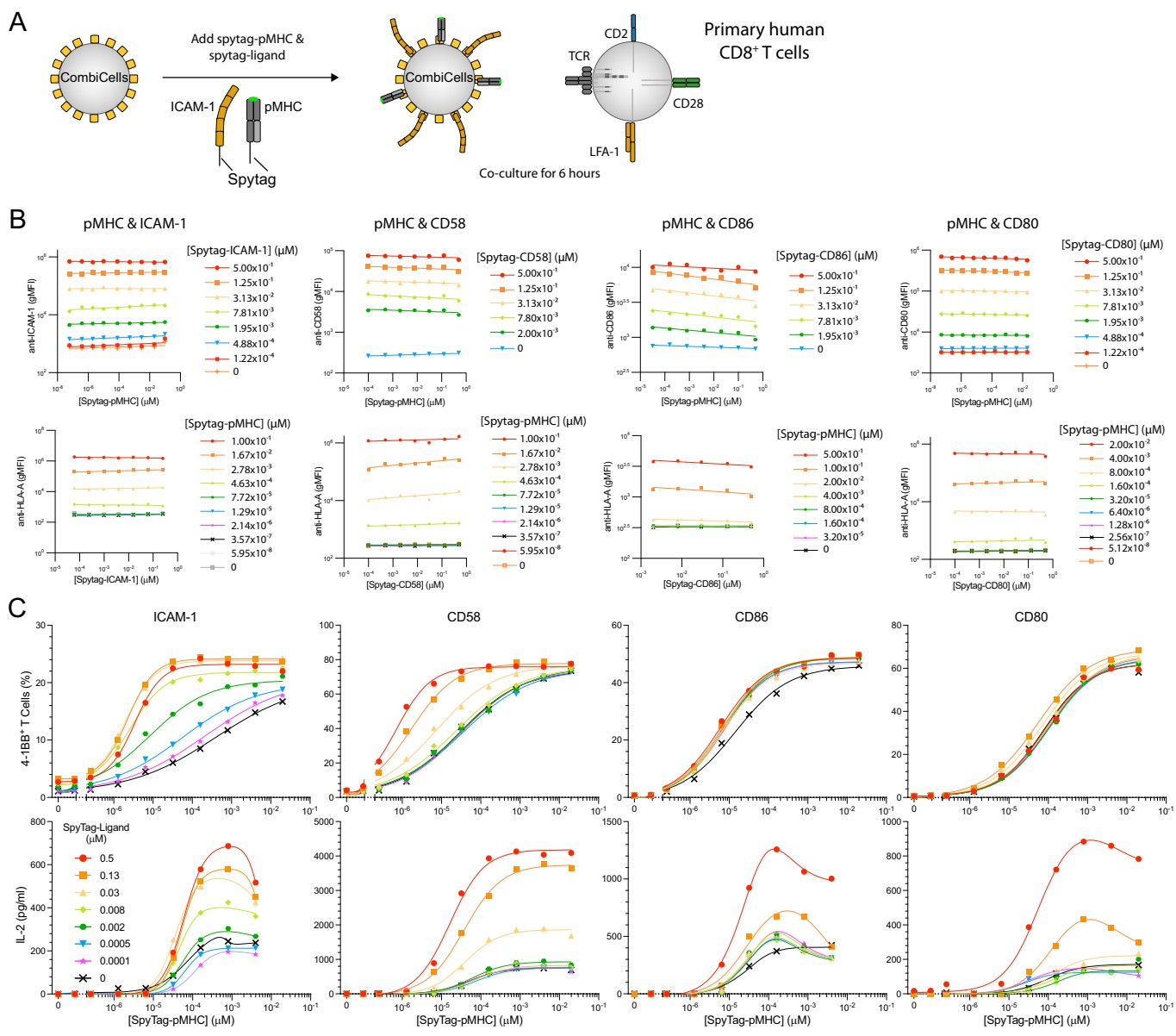

**Figure 2. T cell activation is determined by the combinatorial display of ligands.**

(**A**) Experimental workflow. (**B**) Surface level of the indicated ligand (top row) and pMHC (bottom row) detected by flow cytometry after coupling the indicated combination of ligand and pMHC. (**C**) T cell activation measured by surface 4-1BB (top row) or supernatant IL-2 (bottom row) after 6 h of co-culture with CombiCells (Figure EV1 for additional activation data). Data information: Experiments in each column of (**B**, **C**) were performed independently by transducing primary human CD8 + T cells isolated from different leukocyte cones. Source data are available online for this figure.

CD28 on CAR cytokine production was not unexpected given that it already contained intrinsic CD28 co-stimulation (Fig. 3E).

## The accessory receptor LFA-1 primarily controls the sensitivity of CD19 targeting CARs

We next investigated the antigen sensitivity of two clinically approved CAR-T cell therapies targeting the folded antigen CD19 on the surface of B cells, Yescarta and Kymriah. These second-generation CARs use the same FMC63 recognition domain fused to either the CD28 hinge, transmembrane, and co-stimulation

domains (Yescarta) or CD8 hinge and transmembrane regions and the 4-1BB co-stimulation domains (Kymriah). The current method for studying CAR-T cell antigen sensitivity is to generate panels of cells expressing different levels of the antigen (Haso et al, 2012; Majzner et al, 2020). However, using a panel of the B cell leukemia Nalm6 cell lines, we found T cell activation was already maximal in response to the clone with the lowest CD19 levels, which was barely detectable by flow cytometry (Figure EV4). We also observed this when studying antigen sensitivity by the TCR finding T cell activation at concentrations of pMHC that were lower ($<10^{-5}$ μM, Fig. 2C) than the concentrations required to

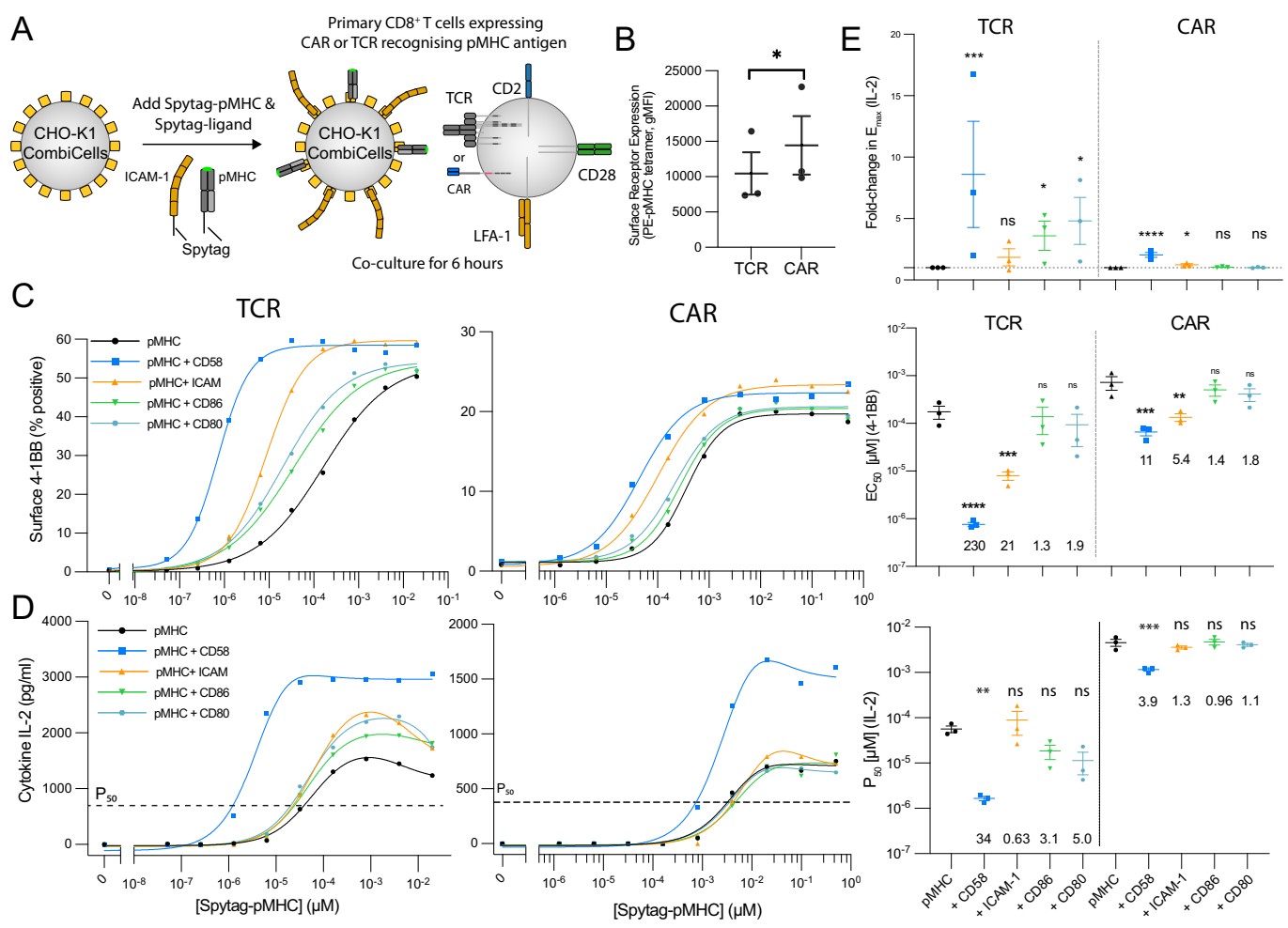

**Figure 3. The TCR is more efficient than the CAR at exploiting CD2 and LFA-1 to increase antigen sensitivity.**

(A) Schematic of assay. (B) Surface antigen receptor assessed using pMHC tetramer from $N = 3$ independent experiments. (C, D) Representative dose-response for the TCR and CAR (left) and summary measures of antigen sensitivity across $N = 3$ independent experiments (right). (E) The fold-change in the maximum IL-2 secreted relative to pMHC alone from the experiments in (D) using $N = 3$ independent experiments. See Fig. EV3 for additional measures of T cell activation. Data information: A paired $t$ test (B) or a $t$ test with Dunnett's multiple comparison correction on log-transformed values (C–E) is used to determine $p$ values. *$p$ value ≤ 0.05, **$p$ value ≤ 0.01, ***$p$ value ≤ 0.001, ****$p$ value ≤ 0.0001. Source data are available online for this figure.

detect pMHC by flow cytometry ($>10^{-4}$ µM, Fig. 2B). This inability to measure antigen surface densities in range needed for measuring T cell sensitivity highlights another advantage of being able to titrate surface antigen levels on cells.

In order to titrate CD19 on target cells, we produced Nalm6 CombiCells by transducing hCD52-Spycatcher into CD19 KO Nalm6 cells (Fig. 4A) and confirmed that purified Spytag-CD19 can readily couple to the cell surface (Fig. 4B). The surface expression of CD19 remained stable for over 24 h on Nalm6 CombiCells with a lifetime of 49 h (Fig. 4C). The surface expression of Spytag-CD19 was less stable on CHO-K1 CombiCells or the U87 glioblastoma cell line expressing hCD52-Spycatcher (Fig. 4C).

When Nalm6 CombiCells were loaded with a range of concentrations of Spytag-CD19 and used to stimulate primary CD8⁺ T cells expressing either CAR, the antigen sensitivity of Yescarta was 6.3 to 11.5-fold higher than Kymriah (Figs. EV5 and 4D). T cell activation, as measured by 4-1BB surface expression, was detected even when CD19 levels on the

Nalm6 surface were too low to detect by flow cytometry (Fig. 4D, black arrow). To investigate the contribution of accessory receptors, we used the CHO-K1 CombiCell assay (Fig. 4E). In contrast to the pMHC-targeting TCR and CAR, we found that the antigen sensitivity of these CD19-targeting CARs was enhanced more by LFA-1 than by CD2 ligands (Fig. 4F). This suggests that CD2 is not being efficiently exploited by the CD19 CARs currently licensed for clinical use.

As observed with the pMHC-targeting CAR (Fig. 3), ligation of accessory receptors had only a modest impact on cytokine production by CD19-targeting CARs (Appendix Fig. S3A,B). This may reflect the fact that these are 2nd generation CARs containing CD28 (Yescarta) or 4-1BB (Kymriah) costimulatory motifs, respectively, which activate co-stimulation pathways. Importantly, however, their antigen sensitivity, as measured by 4-1BB expression and cytokine production, remained lower than that achieved by the TCR on CHO-K1 CombiCells (e.g., Fig. 4F and Appendix Fig. S3A,B vs. Figs. 3 and EV3). As before, coupling of each ligand did

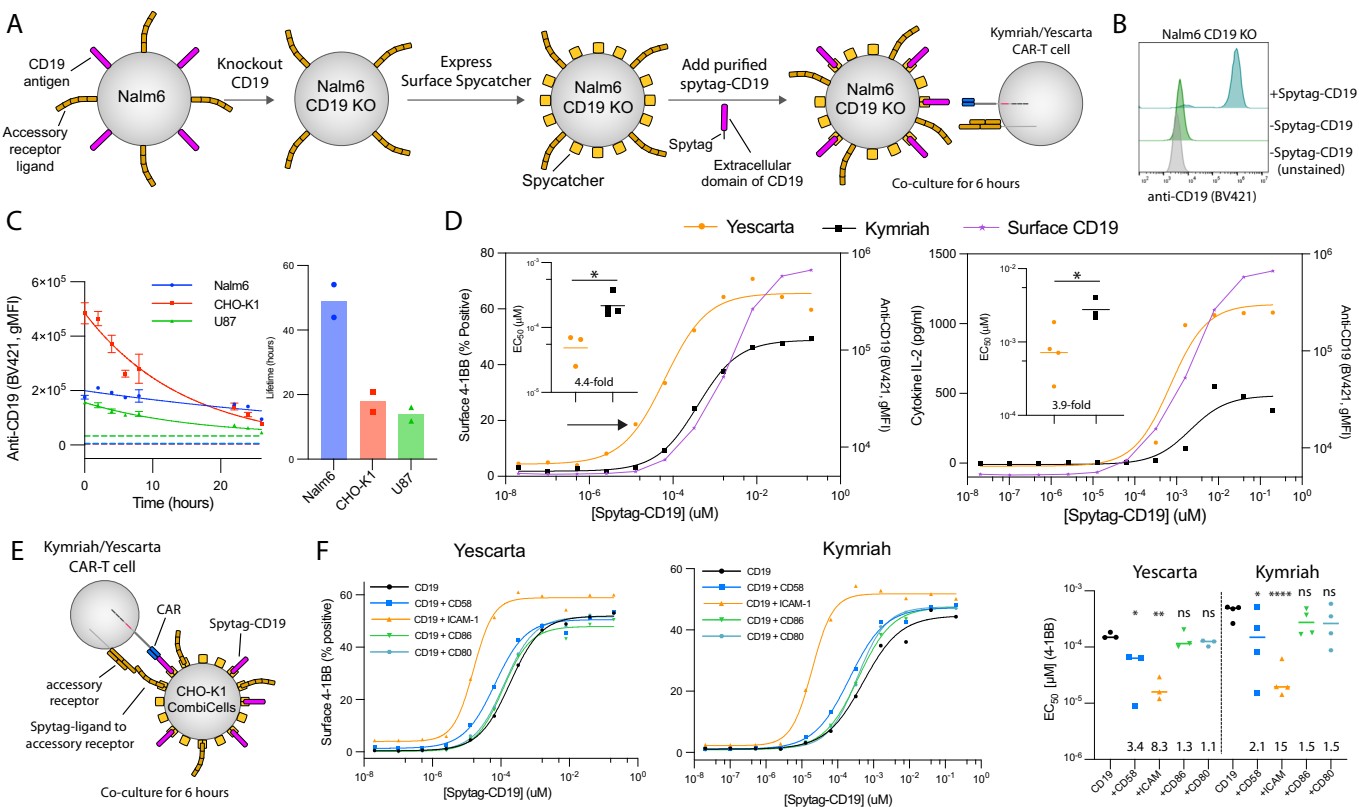

**Figure 4. Yescarta and Kymriah CAR-T cells can exploit LFA-1 but not CD2 or CD28 for improving their antigen sensitivity.**

(A) Schematic of protocol for producing CD19− hCD52-Spycatcher+ Nalm6 cells. (B) Surface level of Spytag-CD19 following coupling to hCD52-Spycatcher. (C) Representative timecourse of surface Spytag-CD19 on the indicated cell lines showing the mean and SD from 3 technical replicates (left) and fitted lifetime from $N = 2$ independent experiments (right). Horizontal dashed lines show unloaded controls. (D) A representative experiment showing T cell activation by 4-1BB (left) and supernatant IL-2 (right) with surface levels of CD19 on the target Nalm6 cell (right y-axes) and summary measures across $N = 3$ (Yescarta) and $N = 4$ (Kymriah) independent experiments (inset). (E) Schematic of CAR-T cell assay recognizing CD19 alone or in combination with ligands to accessory receptors on CHO-K1 CombiCells. (F) A representative experiment showing T cell activation by 4-1BB (left, middle) and summary measures across $N = 3$ (Yescarta) and $N = 4$ (Kymriah) independent experiments (right). Data information: A paired $t$ test (D) or a $t$ test with Dunnett's multiple comparison correction (F) both on logtransformed values is used to determine $p$ values. *$p$ value ≤ 0.05, **$p$ value ≤ 0.01, ***$p$ value ≤ 0.001, ****$p$ value ≤ 0.0001. Source data are available online for this figure.

not impact the coupling of CD19 antigen or vice versa (Appendix Fig. S3D,E). Taken together, this suggests that CARs are inefficient at exploiting accessory receptors to increase their antigen sensitivity.

## The antigen sensitivity of the TCR is higher than CARs and BiTEs

We next used the CD19 KO Nalm6 CombiCells to quantify the antigen sensitivity of Blinatumomab, which is an approved BiTE targeting CD19. By including Kymriah and Yescarata, we determined that Blinatumomab performed better than Kymriah and similar to Yescarta in terms of surface 4-1BB, secreted IL-2, and cytotoxicity (Fig. 5A; Appendix Fig. S4A). We confirmed that Blinatumomab was not limiting in these experiments as higher concentrations did not impact sensitivity (Appendix Fig. S4B).

We noted that the maximum antigen sensitivity of the TCR on CHO-K1 CombiCells (Fig. 3C EC$_{50}$ ≈ 10$^{-6}$ μM with Spytag-CD58 for 4-1BB) was higher than the maximum for the CD19 CARs on CHO-K1 CombiCells (Fig. 4F—EC$_{50}$ > 10$^{-5}$ μM with Spytag-ICAM-1 for 4-1BB). To assess whether this difference is maintained

on Nalm6 cells, we quantified the antigen sensitivity of the 1G4 TCR using $\beta_2$M KO Nalm6 CombiCells (Fig. 5B). These cells were necessary because even minute quantities of free peptide in the Spytag-pMHC preparation could be loaded onto HLA-A*02:01 on the parental Nalm6 line. The antigen sensitivity of the TCR on these $\beta_2$M KO Nalm6 CombiCells was similar to CHO-K1 CombiCells loaded with Spytag-CD58 (Fig. 5B, e.g., EC$_{50}$ ≈ 10$^{-6}$ μM for 4-1BB). This high sensitivity for the TCR was >100-fold and >10-fold larger than the antigen sensitivity of CD19 targeting BiTE/ CARs for 4-1BB and IL-2 secretion, respectively (Fig. 5A, see dashed lines).

Taken together, the native TCR appears to display higher antigen sensitivity compared to Blinatumomab and Yescarta, which are more sensitive than Kymriah.

## The inhibitory PD-1/PD-L1 interaction can inhibit the isolated recognition of pMHC and CD2/CD28 co-stimulation

The accessory receptor PD-1 is known to inhibit T cell activation but it is unclear whether it primarily inhibits TCR signaling, CD28

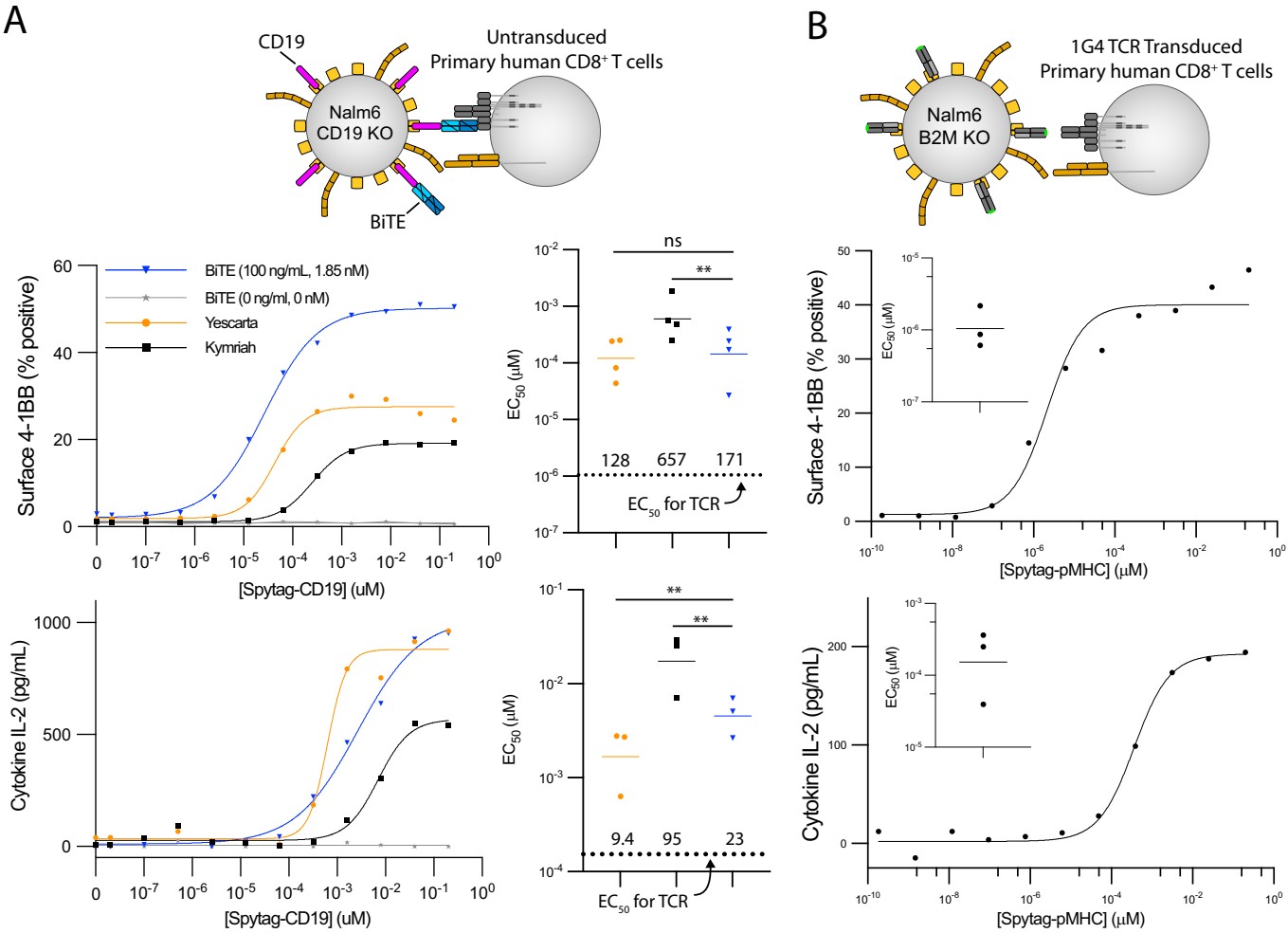

**Figure 5. The antigen sensitivity of the pMHC-targeting TCR is higher than the CD19-targeting Blinatumomab (BiTE) and Yescarta/Kymriah (CARs) when recognizing antigen on Nalm6 CombiCells.**

(A) Untransduced CD8[+] T cells responding to CD19 KO Nalm6 CombiCells loaded with different concentrations of Spytag-CD19 with the indicated concentration of BiTE. Kymriah and Yescarta transduced CD8[+] T cells are included for comparison. A representative experiment (left) and summary $EC_{50}$ values (right) for $N = 3$ independent experiments. (B) 1G4 TCR transduced CD8[+] T cells responding to $\beta_2$M KO Nalm6 CombiCells loaded with Spytag-pMHC. Representative experiment and fitted $EC_{50}$ values for $N = 3$ independent experiments (inset). The mean $EC_{50}$ value from B is shown as a horizontal dotted line in (A). Data information: In (A), a $t$ test with Dunnett's multiple comparison correction is used to determine $p$ values on log-transformed $EC_{50}$ values. **$p$ value $\leq 0.01$. Source data are available online for this figure.

signaling, or both (Celis-Gutierrez et al, 2019; Hui et al, 2017; Kamphorst et al, 2017; Mizuno et al, 2019). Moreover, it is presently unknown whether PD-1 inhibits co-stimulation by other surface receptors, such as CD2. To investigate this, we used CHO-K1 CombiCells to stimulate CD8[+] PD-1[+] 1G4 TCR[+] Jurkat T cells with pMHC alone or with different combinations of ligands to CD28 (CD80), CD2 (CD58) and PD-1 (PD-L1). We used the Jurkat T cell line because it is an established assay for PD-1 function (Celis-Gutierrez et al, 2019; Hui et al, 2017; Kamphorst et al, 2017; Mizuno et al, 2019) and because there is presently no robust assay for PD-1 function in primary human CD8[+] T cells.

As expected, ligands for CD28 or CD2 greatly increased T cell activation by pMHC (Fig. 6A–C). In contrast, the PD-1 ligand abolished T cell activation by TCR ligation alone as well as by simultaneous TCR and CD28 ligation. Interestingly, PD-1 ligation

also abolished T cell activation by simultaneous TCR and CD2 ligation. These results could not be explained by PD-L1 coupling simply displacing pMHC, CD80, or CD58 because their surface levels were not reduced by coupling of PD-L1 (Appendix Figure S5). Lastly, Jurkat T cells produce only a limited number of cytokines compared to primary T cells (Bartelt et al, 2009) and because the Jurkat T cell clone we have used produced only modest levels of IL-2, we have used CD69 and IL-8 and both of these measures of T cell activation produced similar results. Moreover, 4-1BB expression correlates well with CD69 expression and IL-2 secretion correlates well with other cytokines in primary T cells (Figs. 3C–E and EV3). Taken together, these data indicate that PD-1 ligation directly inhibits TCR signaling and therefore, the ability of PD-1 to inhibit CD2 and CD28 co-stimulation may be a result of removing the primary TCR signal and/or its ability to directly inhibit CD2 and CD28 signaling (Fig. 6D).

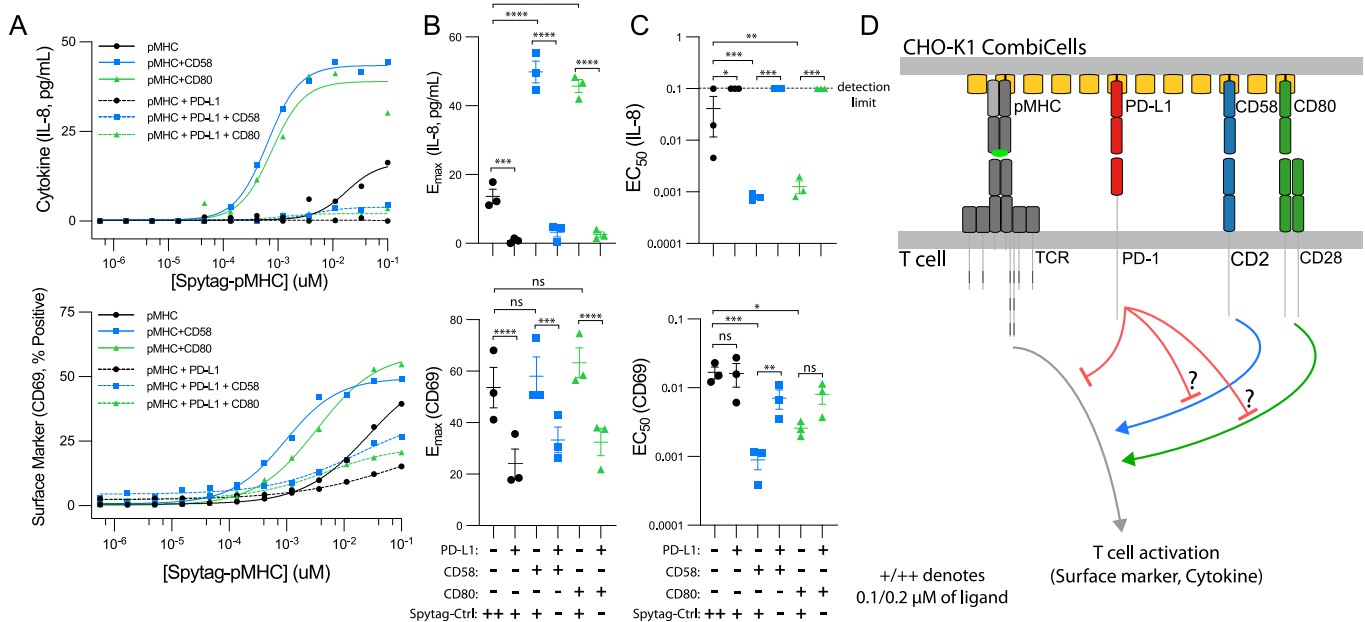

**Figure 6. Ligation of PD-1 inhibits T cell activation in response to pMHC alone or in combination with CD28 and CD2 co-stimulation.**

(A–C) Jurkat T cells transduced to express CD8, CD2, PD-1, and the 1G4 TCR were co-cultured with CHO-K1 CombiCells coupled with the indicated combinations of ligands for 20 h. The protein CD19 fused to Spytag was used as placeholder control protein (Spy-Ctrl) to ensure that the total concentration of Spytag accessory receptor ligands remained constant (see Appendix Fig. S5 for surface levels). (A) Representative dose-response for IL-8 (top) and CD69 (bottom) and (B, C) fitted metrics across $N = 3$ independent experiments. (D) Schematic of proposed inhibition mechanisms of PD-1. Data information: In (B, C), mean and SEM are shown. A $t$ test with Dunnett's multiple comparison correction directly (B) or on log-transformed values (C) is used to determine $p$ values. *$p$ value ≤ 0.05, **$p$ value ≤ 0.01, ***$p$ value ≤ 0.001, ****$p$ value ≤ 0.0001. Source data are available online for this figure.

# Discussion

We have developed a new CombiCell platform for studying cell–cell recognition. It adapts the Spycatcher/Spytag split proteins system by expressing a novel membrane-anchored Spycatcher on the surface of cells selected and engineered to lack ligands under investigation. Soluble ligands fused to a membrane-proximal Spytag can readily be coupled to these cells in different combinations and concentrations. This platform, which we call CombiCells, removes a major bottleneck that has been slowing down studies of cell–cell recognition.

CombiCell has several advantages over existing methods, which typically rely on genetic modifications coupled to cell sorting to produce many cell lines with different concentrations and combination of ligands. Firstly, it greatly reduces the number of cell lines. For example, testing just 12 concentrations of antigen with 4 different ligands (e.g., Fig. 4F) would require an impractical 60 cell lines using current genetic methods. With CombiCells only one cell line is required.

Secondly, cell lines grown independently in culture undergo genetic drift, making it difficult to rule out that observed differences are not the result of such changes. While this could be addressed by creating duplicate cell lines expressing each ligand combination, this would further increase the number of cell lines required. Using CombiCells reduces the time that cells with different ligand combinations are independently cultured from weeks/months in the standard approach to minutes. Finally, titration allows ultra-low levels to be displayed on the target cell

that are impossible to quantify by flow cytometry. This is crucial when measuring highly sensitive recognition, such as by T cells, which can recognize a single antigen on a target cell (Huang et al, 2013; Siller-Farfan and Dushek, 2018).

To exploit CombiCells, we have focused on T cell activation because the infected or cancerous cells targeted by T cells often modulate expression of surface molecules to evade immune recognition. We show that engagement CD2 had a bigger impact than engagement of LFA-1 or CD28 when T cells recognize pMHC antigens using their TCR. In contrast, LFA-1 had the biggest impact when T cells recognized the cancer antigen CD19 using the clinically approved Yescarta and Kymriah CARs. This is consistent with a recent report showing improved CAR-T cell responses when increasing ICAM-1 expression (Larson et al, 2022), and suggests that CARs may be under-utilizing CD2. Consistent with a previous report, we found that Yescarta achieved higher antigen sensitivity compared to Kymriah (Majzner et al, 2020) and we now report that Blinatumomab (BiTE) performs similarly to Yescarta but that the TCR outperforms both by >10-fold. A limitation of these results is that it was not possible to match the surface expression of the different antigen receptors used. While BiTEs can use all surface TCR-CD3, in TCR-transduced T cells, only a subset of TCR-CD3 will have the correct TCR for targeting the pMHC. Furthermore, pMHC-targeting CARs are typically expressed at higher levels than the TCR (Burton et al, 2023), presumably because CAR expression, unlike TCR expression, is not limited by the availability of CD3 subunits. Taken together, this suggests that the TCR's high sensitivity is unlikely to result from higher surface expression compared to BiTEs and CARs.

Recent studies have suggested that the inhibitory effect of PD-1 involves dephosphorylation of the cytoplasmic tail of CD28 (Hui et al, 2017; Xu et al, 2020). We find that PD-1 engagement by PD-L1 can inhibit T cell activation in response to pMHC alone, suggesting that it can also dephosphorylate activatory tyrosines in the TCR signaling pathway (Celis-Gutierrez et al, 2019; Mizuno et al, 2019), as originally proposed (Freeman et al, 2000). We also show that PD-1 can inhibit T cell activation enhanced by costimulation through CD2. While CD2 does not contain any tyrosines in its cytoplasmic tail, it has been shown to recruit the tyrosine-containing activatory kinase Lck (Beyers et al, 1992). Our results, taken together with previous reports, are consistent with a model where PD-1 promiscuously inhibits many pathways involving tyrosine phosphorylation (Boussiotis, 2016; Clemens et al, 2021).

While the CombiCell platform has numerous advantages, it also has limitations. First, because these Spycatcher-coupled ligands lack their native membrane/cytoplasmic domains, ligands whose function is influenced by these domains may behave differently. Second, even when initial surface densities of the native and Spycatcher-coupled ligands are matched, degradation of Spytag-protein/Spycatcher complexes will result in divergence in expression levels during the assay. As a result, additional ligand may be required for assays that are long compared to the half-life Spytag-ligand/Spycatcher complexes, which can range from ≈7 h (Fig. 1H) to >24 h (Fig. 4C). Third, this system is not suitable for capturing ligands with multiple transmembrane domains.

Another limitation of this system is that coupling the ligands and antigen extracellular domains to surface Spycatcher increases their physical size. The C-terminus of coupled ligands could be positioned up to ≈3 nm above the surface (Fig. 1A). Previous work has demonstrated that increasing the TCR ligand size by 7 nm drastically reduces T cell activation (Choudhuri et al, 2005), perhaps by impairing TCR segregation from the receptor tyrosine phosphatase CD45, as proposed in the kinetic-segregation model (Dushek et al, 2012; van der Merwe and Dushek, 2011). The fact that we observed robust T cell activation with high sensitivity (EC$_{50}$ ~$10^{-6}$ μM) when antigen was displayed on surface Spycatcher suggests that the modest elongation of the ligand by the Spytag/Spycatcher system has limited impact on TCR triggering. While it should be possible to reduce the overall dimension of coupled ligands or antigens by truncating their native extracellular hinge/stalk, it has been shown that relatively small (5 nm) size differences can prevent colocalization of receptor/ligand complexes at membrane/membrane interfaces (Schmid et al, 2016). Our approach of coupling the full extracellular domains of all ligands and antigens to surface Spycatcher has the advantage of not introducing size differences.

Different T cell subsets can express different combinations of accessory receptors (Chen and Flies, 2013). While we have focused on using primary human CD8$^+$ T cell blasts, we anticipate that the impact of accessory receptors on antigen sensitivity will be conserved between subsets of T cells that express these receptors. Indeed, CD2 and LFA-1 are known to increase antigen sensitivity of both naive and previously activated T cells (Abu-Shah et al, 2020; Bachmann et al, 1999, 1997; Burton et al, 2023; Pettmann et al, 2023, 2021; Trendel et al, 2021). We focused on CD2 and LFA-1 in our study because previous work has established that they can increase the antigen sensitivity of the TCR (Abu-Shah et al, 2020; Bachmann et al, 1999, 1997; Burton et al, 2023; Pettmann et al,

2023, 2021; Trendel et al, 2021). Although TNFRSF members CD27, 4-1BB, GITR, and OX40 are known to be important co-stimulatory receptors on T cells they appear to have only a modest impact on antigen sensitivity (Nguyen et al, 2021; Trendel et al, 2021). Expanding this analysis to a broader set of accessory receptors in diverse T cell subsets would be expedited by using Combicells.

By introducing CombiCells we have provided a platform that greatly facilitates the study of receptor/ligand interactions at cell/cell interfaces. We have utilized the platform to compare antigen sensitivity of TCRs, CARs, and BiTEs and the contribution of various accessory receptors to T cell activation, including an inhibitory receptor. This platform can be deployed to examine higher-order combinations of ligands, other surface receptors, and different cell types. CombiCells enable analysis of ligand/receptor interactions at cell/cell interfaces with the convenience hitherto restricted to those studying soluble ligands. This platform should enhance our understanding of how cells integrate signals from diverse surface receptor/ligand interactions at cell–cell interfaces.

# Methods

## Protein production and purification

### Production of Spytag-pMHC

HLA-A*02:01 heavy chain (UniProt residues 25–298) with a C-terminal Spytag003 and β$_2$-microglobulin were expressed as inclusion bodies in *E. coli*, refolded in vitro as described in (Aleksic et al, 2010) together with the 9 V NY-ESO-1 peptide, and purified using size-exclusion chromatography on a Superdex S75 column (GE Healthcare, USA) in HBS-EP buffer (10 mM M HEPES pH 7.4, 150 mM NaCl, 3 mM EDTA, 0.005% v/v Tween-20).

All ligands contained the full extracellular domain fused to a c-terminal Spytag003 (RGVPHIVMVDAYKRYK) followed by a Histag for purification (HHHHHH).

### Production of Spytag-ICAM-1/CD58/CD86/CD80/PD-L1/CD19

Expi293™ cells (ThermoFisher Scientific, A14527) were grown in Expi293™ Expression Medium (ThermoFisher Scientific, A1435101) in a 37 °C incubator with 8% CO$_2$ on a shaking platform at 130 rpm. Cells were passaged every 2–3 days with the suspension volume always kept below 33.3% of the total flask capacity. The cell density was kept between 0.5 and 3 million per ml. Before transfection cells were counted to check that cell viability was above 95%, and the density was adjusted to 3.0 million per ml. For 100 ml transfection, 320 μl ExpiFectamine™ 293 Transfection reagent (ThermoFisher Scientific, A14524) was mixed with 6 ml Opti-MEM (ThermoFisher Scientific, 31985062) for 5 min. During this incubation, 100 μg of expression plasmid was mixed with 6 ml Opti-MEM. The DNA was then mixed with the ExpiFectamine™ and incubated for 15 min before being added to the cell culture. One day after transfection 600 μl of enhancer 1 and 6 ml of enhancer 2 was added to the culture flask. The culture was returned to the shaking incubator for 4–5 days for protein expression to take place.

Cells were harvested by centrifugation and the supernatant collected and filtered through a 0.22 μm filter. Imidazole was added

to a final concentration of 1 mM and PMSF added to a final concentration of 1 mM; 2 ml of Ni-NTA Agarose (Qiagen, 30310) was added per 50 ml of supernatant and the mix was left on a rolling platform at 4 °C overnight. The mix was poured through a gravity flow column to collect the Ni-NTA Agarose. The Ni-NTA Agarose was washed three times with 10 ml of wash buffer (50 mM NaH2PO4, 300 mM NaCl, and 5 mM imidazole at pH 8). The protein was eluted with 15 ml of elution buffer (50 mM NaH2PO4, 300 mM NaCl, and 250 mM imidazole at pH 8). The protein was concentrated, and buffer exchanged into size exclusion buffer (25 mM NaH2PO4 and 150 mM NaCl at pH 7.5) using a protein concentrator with a 10,000 molecular weight cut-off. The protein was concentrated down to 500 µl and loaded onto a Superdex 200 10/300 GL (Cytiva, 17-5175-01) size exclusion column. Fractions corresponding to the desired peak were pooled and frozen at −80 °C. Samples from all observed peaks were analyzed on a reducing SDS–PAGE gel.

For purified Spytag-CD19, SUMO was used to stabilize the protein during production and therefore the HRV 3C Protease Solution Kit was used for SUMO removal (Pierce™, 88946). HRV protease was added to the purified protein at a pre-determined optimum ratio for full cleavage of the HRV site. The mixture was left overnight for full cleave to occur and then 1 ml of Glutathione Agarose (Pierce™, 16100) added for 4 h to remove the protease. The solution was run through a gravity flow column to collect to SUMO plus protein of interest mixture. This was then added to 1 ml of Ni-NTA Agarose (Qiagen, 30310) and left on a rolling platform at 4 °C overnight. The mix was poured through a gravity flow column to collect the Ni-NTA Agarose. The Ni-NTA Agarose was washed once with 10 ml of wash buffer (50 mM NaH2PO4, 300 mM NaCl, and 5 mM imidazole at pH 8). The protein was eluted with 15 ml of elution buffer (50 mM NaH2PO4, 300 mM NaCl, and 250 mM imidazole at pH 8). The protein was concentrated, and buffer exchanged into size exclusion buffer (25 mM NaH2PO4 and 150 mM NaCl at pH 7.5) using a protein concentrator with a 10,000 molecular weight cut-off and frozen in suitable aliquots at −80 °C.

## Generation of ICAM-1 knockout CHO-K1 cells

The expression of the hamster surface molecule ICAM1 was eliminated on CHO-K1 cells (ATCC CCL-61) using CRISPR/Cas9 lipofection, followed by lentiviral introduction of surface Spycatcher with the human CD52 hinge. Cells were maintained in DMEM (Sigma Aldrich) with 10% FCS (Sigma Aldrich). First, 200,000 were seeded overnight in a 6-well plate, followed by transfection with Lipofectamine CRISPRMAX Cas (Invitrogen), TrueCut Cas9 Protein v2 (Invitrogen), and an ICAM1 exon 2 (Ig domain 1)-targeting TrueGuide sgRNA (Invitrogen; sequence: CCACAGTTCTCAAAGCACAG) according to the manufacturer's U2OS protocol. Specifically, 125 µl OptiMEM (Thermo Fisher), 6.25 µg (37.5 pmol) Cas9, 3.75 µl of 10 µM sgRNA in TE (37.5 pmol), and 2.5 µl Lipofectamine Cas9 Plus were mixed in one tube. Separately, 125 µl OptiMEM, and 7.5 µl Lipofectamine CRISPRMAX were mixed and incubated for 1 min. Both tubes were combined and incubated for 15 min at RT. Finally, 50 µl of the solution was added per well of CHO cells. After 1 week, single clones were grown by performing limiting dilution.

Clones were screened using Sanger sequencing after genomic PCR. Specifically, gDNA from outgrown single cell clones was isolated using PureLink Genomic DNA Mini Kit (Invitrogen), amplified in a PCR with fwd primer AGGCATCAGATGGTGG-CATTCT and rev primer GGTGTTTGGGGAGGGCAATACT, and submitted for Sanger sequencing. A clone which showed genomic editing was selected for further processing. Next, surface Spycatcher was introduced using high MOI lentiviral transduction, followed by single cell cloning using limiting dilution. The final clone selected showed high expression of surface SpyCatcher and absence of ICAM1 on the cell surface by flow cytometry. The expression of surface Spycatcher was assessed by coupling purified Spytag-mClover and flow cytometry. Specifically, 100k cells were incubated with 10 µM Spytag-mClover in PBS for 1 h at RT in the dark, washed in PBS, and acquired on a flow cytometer. ICAM1 expression was tested using unpurified Y5-3F9 hybridoma super-natant (provided by Vijay Kuchroo and Edward Greenfield). 100,000 cells were incubated with undiluted Y5 supernatant for 30 min on ice in the dark. Cells were washed in PBS and stained with 1:200 anti-mouse Alexa Fluor-488 secondary antibody for 30 min on ice in the dark. Finally, cells were washed and acquired on a flow cytometer.

## sFCS measurements of diffusion

In all, $10^5$ CHO-K1 cells expressing surface Spycatcher with different hinges were seeded in 8-well chambered coverslips (µ-Slide 1.5H, ibidi) overnight followed by labeling with 50 nM SpyTag-mClover3 for 30 min at 37 °C. Cells were washed 2× in PBS and imaged in complete medium. Imaging was performed on a Zeiss LSM 780 inverted confocal microscope (Carl Zeiss) equipped with a ×40 C-Apochromat NA 1.2 W FCS objective. mClover3 fluorescence was excited with a 488 nm Argon laser and collected onto hybrid GaAsP detectors (Channel S) using a 488 MBS with the pinhole set to 1 AU. The size of the observation area was calibrated using point-FCS measurements of a dye solution (Alexa Fluor 488, 20 nM) with a known diffusion coefficient (Petrasek and Schwille, 2008), yielding an average $\omega$ of 214 nm. Diffusion coefficients ($D$) were then calculated using the equation $\omega^2 = D \times 4 \times t_{xy}$ where $t_{xy}$ is the transit time. Line-scan FCS was performed by switching the ChS to photon-counting mode and data were collected at the basal cell membrane by acquiring a 52-pixel line (digital zoom ×40) at maximum scanning speed for $10^5$ cycles. Files were saved as .lsm5 files and correlated externally using open-source FoCuS software (Waithe et al, 2018).

## Production of TCR or CAR transduced primary human CD8 + T cells

HEK 293T cells were seeded in DMEM supplemented with 10% FBS and 1% penicilin/streptomycin in 6-well plates to reach 60–80% confluency on the following day. Cells were transfected with 0.25 pRSVRev (Addgene, 12253), 0.53 µg pMDLg/pRRE (Addgene, 12251), 0.35 µg pMD2.G (Addgene, 12259), and 0.8 µg of transfer plasmid using 5.8 X-tremeGENE HP (Roche). Media was replaced after 16 h and supernatant harvested after a further 24 h by filtering through a 0.45 µm cellulose acetate filter. Supernatant from one well of a 6-well plate was used to transduce 1 million T cells.

Human CD8 + T cells were isolated from leukocyte cones purchased from the National Health Service's (UK) Blood and

Transplantation service. This project has been approved by the Medical Sciences InterDivisional Research Ethics Committee of the University of Oxford (R51997/RE001), and all leukocyte cones were anonymised by the NHS before purchase. Isolation was performed using negative selection. Briefly, blood samples were incubated with Rosette-Sep Human CD8+ enrichment cocktail (Stemcell) at 150 µl/ml for 20 min. This was followed by a 3.1-fold dilution with PBS before layering on Ficoll Paque Plus (GE) at a 0.8:1.0 ficoll to sample ratio. Ficoll-Sample preparation was spun at $1200 \times g$ for 20 min at room temperature. Buffy coats were collected, washed and isolated cells counted. Cells were resuspended in complete RMPI (RPMI supplemented with 10% v/v FBS, 100 penicillin, 100 streptomycin) with 50U of IL-2 (PeproTech) and CD3/CD28 Human T-activator Dynabeads (Thermo Fisher) at a 1:1 bead to cell ratio. At all times isolated human CD8 + T cells were cultured at 37 °C and 5% $CO_2$. 1 million T cells in 1 ml of media were subsequently transduced on the following day using lentivirus encoding for the 1G4 TCR, Kymriah CAR, or Yescarta CAR, per the section on lentiviral transduction. On days 2 and 4 post-transduction, 1 ml of media was exchanged and IL-2 was added to a final concentration of 50U. Dynabeads were magnetically removed on day 5 post-transduction. When using the TCR, T cells were further cultured at a density of 1 million/ml and supplemented with 50U IL-2 every other day. When using CARs, T cells were further cultured at a density of 0.5 million/ml and supplemented with 100U IL-2 every other day. T cells were used between 10 and 16 days after transduction.

## Production of Jurkat T cell line

The previously described $TRAC^{(-/-)}TRBC^{(-/-)}$ E6.1 Jurkat T cells (Chen et al, 2021) were successively transduced and sorted with lentivirus for the (i) 1G4 TCR, (ii) human CD8$\alpha$-P2A-CD8$\beta$, (iii) human CD2, and (iv) human PD-1. Each transduction used 2 ml of crude lentivirus supernatant on $1 \times 10^6$ Jurkat T cells and cells were allowed to rest for 48–96 h before being subjected to further operations. Jurkat T cells were sorted through FACS to obtain a highly enriched (>99.5%) population of CD8+ PD-1+ 1G4 TCR+ CD2++ Jurkat T cells.

## Coupling of ligands to CHO-K1 cells

In all, 50,000 CHO cells were seeded in a TC-coated 96-well flat-bottom plate and incubated overnight at 37 °C, 10% $CO_2$. Spytag ligands were diluted to the required concentration in complete DMEM (10% FCS, 1% Penicillin–Streptomycin). Existing media was then removed from CHOs and diluted ligands added in a volume of 50 µl, and incubated for 40 or 60 min at 37 °C, 10% $CO_2$. CHOs were then washed twice with complete DMEM.

## Coupling of ligands to Nalm6 cells

In all, 30,000 Nalm6 cells were seeded in a TC-coated 96-well round bottom plate and incubated overnight at 37 °C, 5% $CO_2$. On experiment day, Nalm6 cells were transferred into a TC-coated 96-well V-bottom plate and spun down for 5 min at $520 \times g$. Spytag ligands were diluted to required concentration in complete RPMI (10%FCS, 1% Penicillin–Streptomycin). Existing media was removed from the Nalm6 cells and the diluted ligands added in a

volume of 50 µl, and incubated for 40 min at 37 °C, 5% $CO_2$. Nalm6 cells were then washed twice with complete RPMI.

## Co-culture assays with TCR or CAR transduced T cells

T cells were counted, and washed once in complete RPMI. In all, 50,000 T cells in 200 µl complete RPMI were added to CHO cells coupled with ligand in a 96-well flat-bottomed plate or to Nalm6 cells coupled with ligands and transferred into a 96-well round-bottomed plate. The cells were spun at $50 \times g$ for 1 min to ensure the T cells settle to the bottom of the plate and make contact with adherent CHO cells. The cells were then incubated at 37 °C, 5% $CO_2$ for 6 h (primary T cells) or 20 h (Jurkat T cells).

## Co-culture assays with untransduced T cells and BiTEs

Blinatumomab (BiTE, InvivoGen cat no. bimab-hcd19cd3) was resuspended in to a concentration of 100 µg/ml in sterile water and stored in single use aliquots at −20 °C until the day of the experiment. Following the coupling of Spytag-CD19, the CD19 KO Nalm6 CombiCells were washed once in media and then seeded at 50,000 cells in 90 µl in 96 well plates. To this, 20 µl of BiTE solution at twice the final concentration (BiTEs were diluted in media) was added and the cells incubated for 30 min at 37 °C. Subsequently, 90 µl of untransduced CD8+ T cells (50,000 cells) were added to give the final indicated BiTE concentration. Effector and target cells were co-cultured for 6 h. Control cells containing only effector, only target and no BiTE conditions were also seeded in the same volume. Forty-five minutes before the end of the co-culture 10× cell lysis solution was added to control wells at the appropriate volume to give a final 1× solution, the corresponding volume of sterile water was added to volume correction wells, both for the subsequent cytotoxicity assay. After 6 h, plates were spun briefly at $50 \times g$ for 3 min and 100 µl of supernatant carefully removed. Fifty µl of the supernatant was used immediately in an LDH release assay using Invitrogen CyQUANT™ LDH Cytotoxicity Assay kits and following the manufacturer's protocol. The remaining supernatant was either used immediately or stored at −20 °C for subsequent cytokine detection (see below).

## Flow cytometry—detection of ligands

Straight after ligand coupling and subsequent washing, 10 mM EDTA was added to the CHO cells to detach them. The cells were transferred to a v-bottom plate and spun for 5 minutes at $500 \times g$, 4 °C. The cells were washed once with PBS–BSA 1% for 5 min at $500 \times g$, 4 °C. To detect ligands, fluorescently conjugated antibodies against proteins of interest were diluted in PBS-BSA (1%), at a 1:200 dilution and added at a volume of 50 µl to CHO cells. The cells were resuspended and incubated for 20 min at 4 °C in the dark. The cells were washed twice in PBS, and resuspended in 75 µl PBS, before running on a flow cytometer.

## Flow cytometry—detection of T cell activation

At the end of the stimulation assay, the supernatant was carefully removed and saved for ELISA analysis. 10 mM EDTA in PBS was then added to detach the T cells and CHOs. The cells were then aspirated and transferred to a v-bottom plate and washed once in

200 µl PBS 1% BSA (500 × *g*, 4 °C, 5 min). Antibodies against T cell activation markers were diluted in PBS 1% BSA at a 1:200 dilution. An anti-CD45 antibody was used to selectively stain T cells and distinguish them from CHO cells during flow cytometry analysis. To detect TCR/CAR expression fluorescently-conjugated peptide-MHC tetramers were added to the staining antibodies at a 1:1000 dilution. A viability dye was also added at a dilution if 1:2500 to distinguish live cells from dead cells. 50 µl of this staining solution was to the cells, before incubating them for 20 min at 4 °C in the dark. The cells were washed twice in PBS, and resuspended in 75 µl PBS, before running on a flow cytometer. Flow cytometry data was analyzed using FlowJo (BD Biosciences).

## Cytokine detection

IL-2 Human uncoated ELISA kit, TNF-*α* Human uncoated ELISA kit, IFN-*γ* Human uncoated ELISA kit, or IL-8 Human uncoated ELISA kit and Nunc MaxiSorp 96-well plates were used according to the manufacturer's instructions. The supernatant from stimulation assays were either undiluted (IL-8) or diluted (all other cytokines) prior to ELISAs. The absorbance at 450 nm and 570 nm were measured using a SpectraMax M5 plate reader (Molecular Devices).

## Reagent list

Expi293™: Expression Medium ThermoFisher Scientific Lot: A1435101

ExpiFectamine™ 293 Transfection reagent ThermoFisher Scientific Lot: A14524

Opti-MEM ThermoFisher Scientific Lot: 31985062

Ni-NTA Agarose Qiagen lot: 30310
Glutathione Agarose Pierce™ Lot: 16100
Superdex 200 10/300 GL Cytiva Lot: 17-5175-01
HRV 3C Protease Solution Kit Pierce™ Lot: 88946
DMEM: Thermo Scientific Lot: 41966029 RPMI
Thermo Scientific: Lot: 21875034 EDTA:
UltraPure 0.5 M EDTA, pH 8.0: Invitrogen Lot: 15-575-020
BSA: Sigma-Aldrich Lot: A7906-500G
IL-2 Human uncoated ELISA kit: Invitrogen Lot: 88-7025-77
TNF-*α* Human uncoated ELISA kit: Invitrogen Lot: 88-7346-77
IFN-*γ* Human uncoated ELISA kit: Invitrogen Lot: 88-7316-77
Nunc MaxiSorp 96-well plates Thermo Fisher Scientific Lot: 442404
Antibodies for Flow Cytometry (all from BioLegend):
CD58 Clone: TS2/9 Fluorophore: APC Catalog: 330918
ICAM-1 Clone: HCD54 Fluorophore: AF647 Catalog: 353114
CD86 Clone: Fluorophore: FITC Catalog: 374203
CD80 Clone: 2D10 Fluorophore: BV421 Catalog: 305221
HLA-A2 Clone: BB7.2 Fluorophore: PE Catalog: 343306
CD69 Clone: FN50 Fluorophore: AF488 Catalog: 310916
4-1BB Clone: 4B4-1 Fluorophore: AF647 Catalog: 309824
CD45 Clone: HI30 Flurophore: BV510 Catalog: 304036
Zombie NIR Fixable Viability Kit Catalog: 423105

## Data analysis

$EC_{50}$ is calculated as the concentration of antigen required to elicit 50% of the maximum response determined for each condition individually, whereas $P_X$ is calculated as the concentration of antigen required to elicit $X$% of the maximum activation determined by the pMHC alone condition.

## Surface Spycatcher sequences

Format: IgK signal sequence-Spycacher003 sequence-flexible linker sequence-extracellular hinge sequence-transmembrane sequence-cytoplasmic sequence

mCD80 (pHR-SIN-BXSpyCatcher003-mCD80):
METDTLLLWVLLLWVPGSTGD-
VTTLSGLSGEQGPSGDMTTEEDSATHIKFSKRDEDGRELAGATMELRDSSGKTISTWISDGHVKDFYLYPGKYTFVETAAPDGYEVATPIEFTVNED
GQVTVDGEATEGDAHT-GSSGSGGSHVSEDFTWEKPPEDPPDSKN-TLVLFGAGFGAVITVVVIVVII-
KCFCKHRSCFRRNEASRETNNSLTFGPEEALAEQTVFL

mCD80-Short (pHR-SIN-BXSpyCatcher003-(short)mCD80, Addgene: 210567):
METDTLLLWVLLLWVPGSTGD-
VTTLSGLSGEQGPSGDMTTEEDSATHIKFSKRDEDGRELAGATMELRDSSGKTISTWISDGHVKDFYLYPGKYTFVETAAPDGYEVATPIEFTVNED
GQVTVDGEATEGDAHT-GSSGSGGS- PPDSKNTLVLFGAGFGAVITVVVIVVII-KCFCKHRSCFRRNEASRETNNSLTFGPEEALAEQTVFL

hCD52 (pHR-SIN-BXSpyCatcher003-hCD52, Addgene: 210565):
METDTLLLWVLLLWVPGSTGD-
VTTLSGLSGEQGPSGDMTTEEDSATHIKFSKRDEDGRELAGATMELRDSSGKTISTWISDGHVKDFYLYPGKYTFVETAAPDGYEVATPIEFTVNED
GQVTVDGEATEGDAHT-GSSGSGGS- TSQTSSP-
SASSNISGGIFLFFVANAIIHLFCFS (TSQTSSPS remains in the mature protein)

## Data availability

This study includes no data deposited in external repositories.

## Peer review information

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

## Acknowledgements

We thank Mark Howarth and Marion H. Brown for helpful discussions on the Spycatcher/Spytag system. We thank Vijay Kuchroo and Edward Greenfield for providing the anti-hamster ICAM-1 antibody. We thank Crystal Mackall and Robbie Majzner for providing the panel of Nalm6 cell lines with different levels of CD19. We thank Simon J. Davis for providing the TCR⁻ Jurkat T cells.

## Author contributions

**Ashna Patel**: Data curation; Formal analysis; Investigation; Visualization; Methodology; Writing—review and editing. **Violaine Andre**: Data curation; Formal analysis; Investigation; Visualization; Methodology; Writing—review and editing. **Sofia Bustamante Eguiguren**: Data curation; Formal analysis; Investigation; Visualization; Methodology; Writing—review and editing. **Michael I Barton**: Data curation; Formal analysis; Investigation; Visualization; Methodology; Writing—review and editing. **Jake Burton**: Data curation; Formal analysis; Investigation; Visualization; Methodology; Writing—review and editing. **Eleanor M Denham**: Data curation; Formal analysis; Investigation; Visualization; Methodology; Writing—review and editing. **Johannes Pettmann**: Data curation; Formal analysis; Investigation; Visualization; Methodology; Writing—review and editing. **Alexander M Mørch**: Data curation; Formal analysis; Investigation; Visualization; Methodology; Writing—review and editing. **Mikhail A Kutuzov**: Investigation. **Jesús A Siller-Farfán**: Methodology; Writing—review and editing. **Michael L Dustin**: Supervision; Investigation; Writing—review and editing. **P Anton van der Merwe**: Supervision; Investigation; Writing—review and editing. **Omer Dushek**: Conceptualization; Data curation; Formal analysis; Supervision; Funding acquisition; Investigation; Visualization; Writing—original draft; Project administration; Writing—review and editing.

## Funding

The work was funded by a Wellcome Trust Senior Fellowship in Basic Biomedical Sciences (207537/Z/17/Z to OD), by a Biotechnology and Biological Sciences Research Council (BBSRC) studentship (BB/M011224/1 to AP), by a Wellcome Trust PhD Studentship in Science (108869/Z/15/Z to AMM), and by a Kennedy Trust for Rheumatology Research Professorship (to MLD).

## Disclosure and competing interests statement

AP, ED, JP, PAvdM, and OD have financial interests in a filed patent application related to CombiCells.

# Expanded View Figures

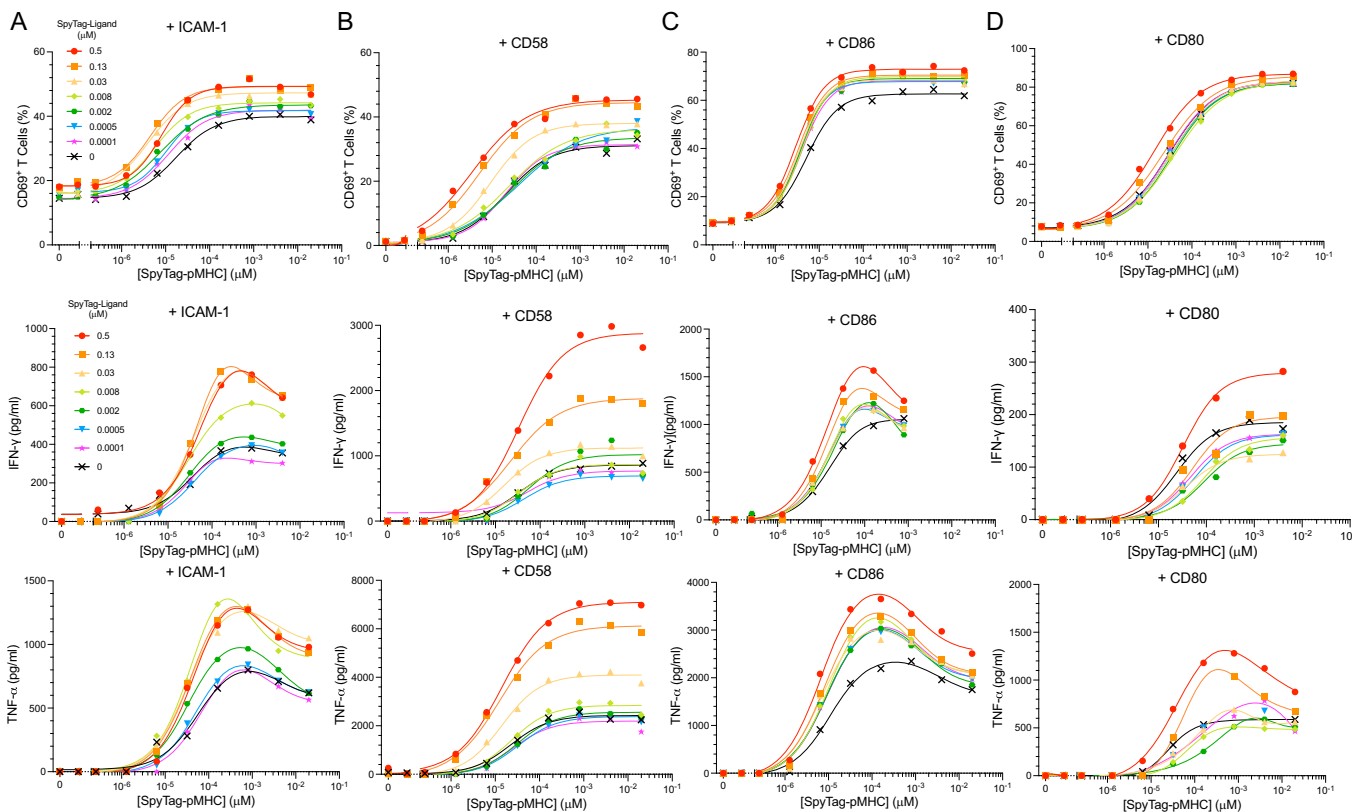

**Figure EV1.   Additional measures of T cell activation.**

Additional measures of T cell activation when coupling (**A**) ICAM-1, (**B**) CD58, **C** CD86, or (**D**) CD80 on CHO-K1 CombiCells along with pMHC (related to Fig. 2).

A

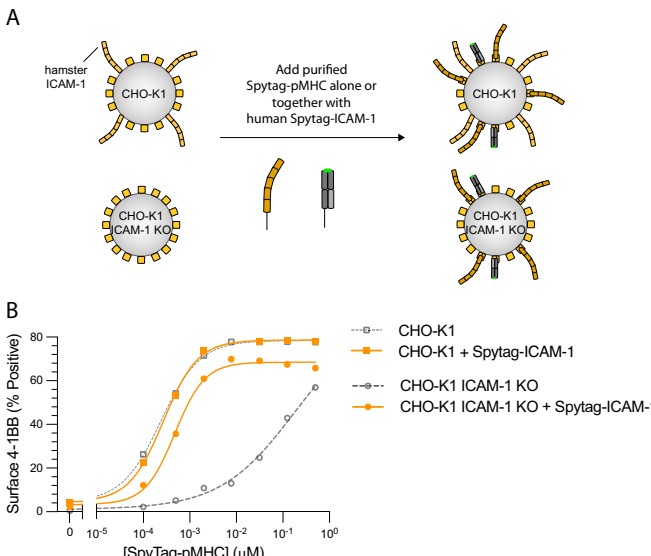

B

Figure legend (chart):

- CHO-K1
- CHO-K1 + Spytag-ICAM-1
- CHO-K1 ICAM-1 KO
- CHO-K1 ICAM-1 KO + Spytag-ICAM-1

Y-axis: Surface 4-1BB (% Positive)

X-axis: [SpyTag-pMHC] (µM)

**Figure EV2. T cells can exploit endogenously expressed hamster ICAM-1 or exogenous human SpytagICAM-1 when recognizing Spytag-pMHC.**

(A) Schematic of CHO-K1 cell lines used. (B) T cell activation measured by the surface marker 4-1BB in response to Spytag-pMHC alone or in combination with 0.5 µM of Spytag-ICAM-1 on the indicated CHO-K1 cell line.

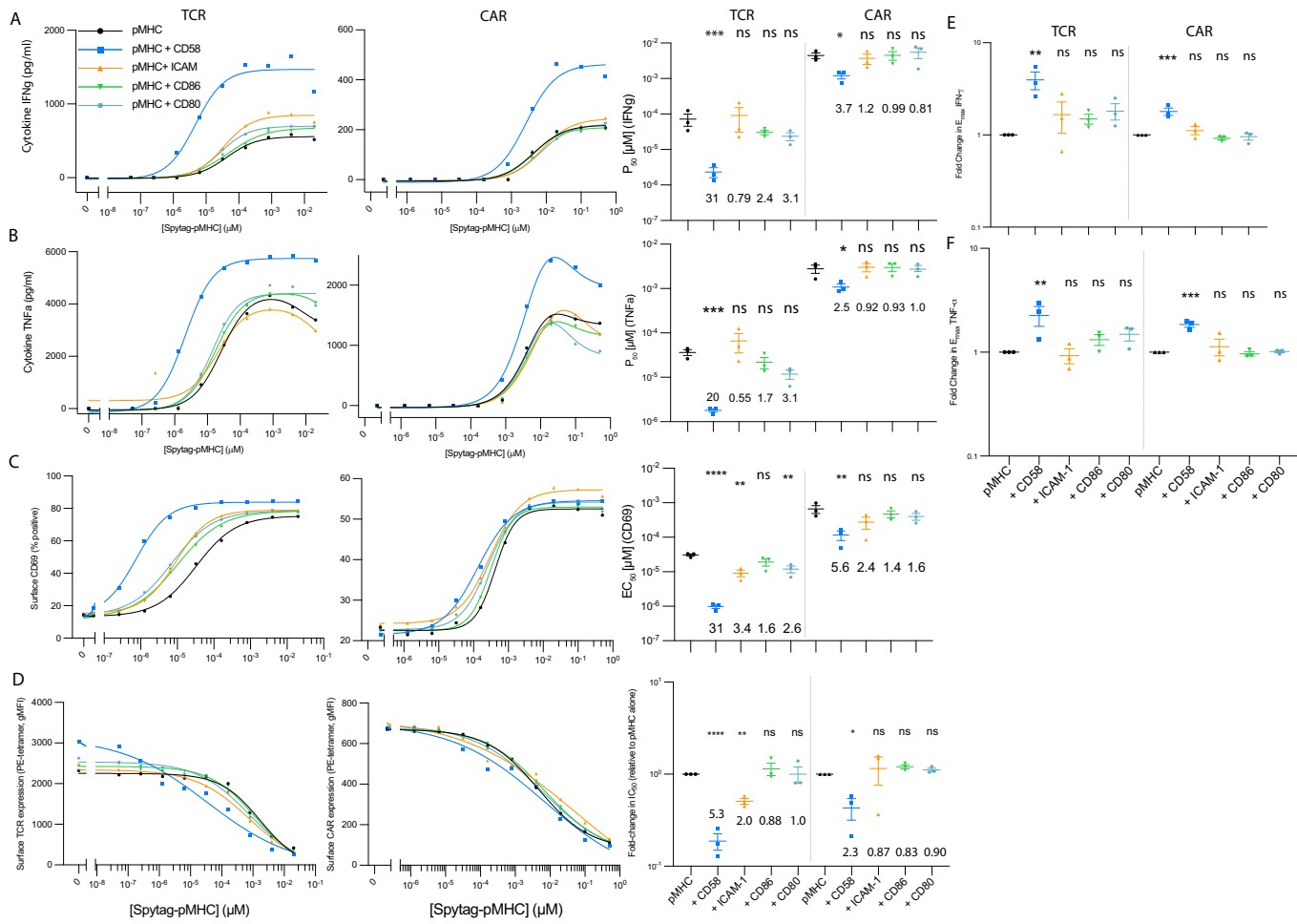

**Figure EV3. Additional measures of T cell activation (related to Fig. 3).**

(A–D) Representative dose-response for the TCR and CAR (left) and summary measures of antigen sensitivity for $N = 3$ independent experiments (right) for (A) IFN-γ, (B) TNF-α, (C) surface CD69, and (D) surface antigen receptor. (E, F) The fold-change in the maximum (E) IFN-γ and (F) TNF-α relative to pMHC antigen alone from $N = 3$ independent experiments. Data information: In (A–D, right panels) and (E) the mean and SEM are shown. A $t$ test with Dunnett's multiple comparison correction on log-transformed values is used to determine $p$ values. *$p$ value ≤ 0.05, **$p$ value ≤ 0.01, ***$p$ value ≤ 0.001, ****$p$ value ≤ 0.0001.

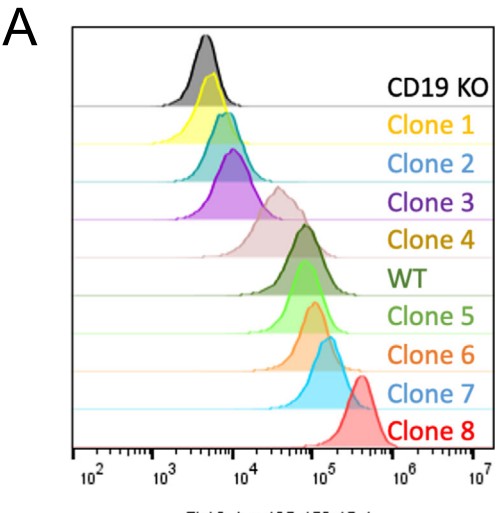

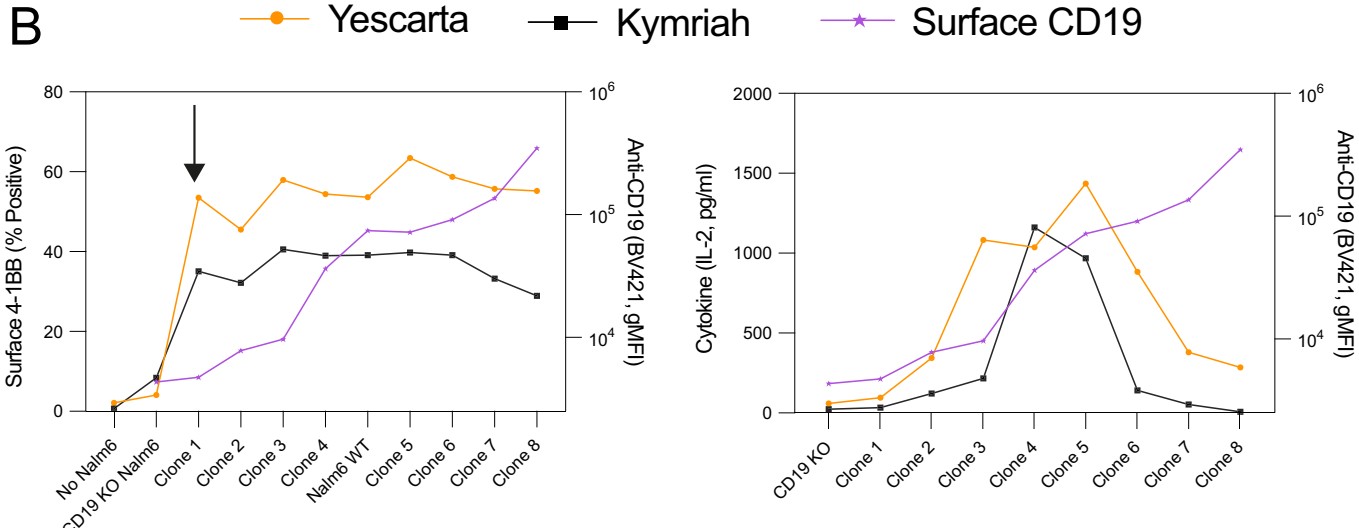

**Figure EV4.  CAR-T cells recognizing CD19 endogenously expressed at different levels on the surface of a panel of Nalm6 cell lines.**

(A) Surface expression of CD19 on the indicated Nalm6 clone. (B) Primary human CD8[+] T cells were co-cultured with the indicated Nalm6 clone for 6 h before T cell activation was assessed by surface 4-1BB (left) and the supernatant levels of IL-2 (right). The complete activation of 4-1BB is observed in response to the Nalm6 cell line expressing the lowest level of CD19 (Clone 1, see right *y*-axes for CD19 level on each Nalm6 cell line). Data information: A representative experiment out of 2 independent experiments is shown.

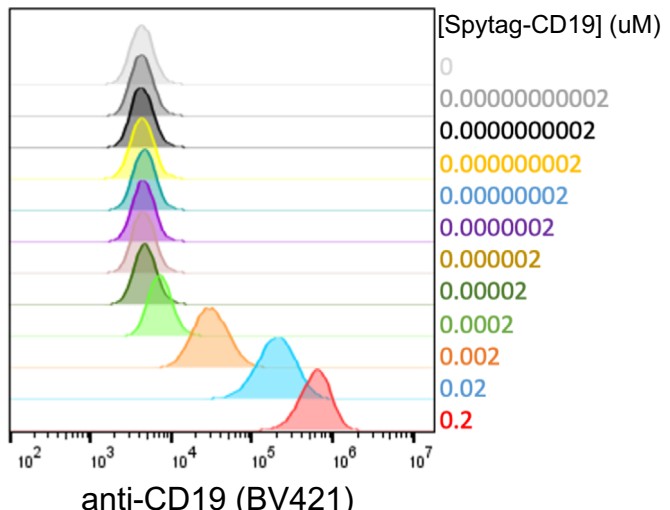

[Spytag-CD19] (uM)
0
0.00000000002
**0.0000000002**
0.000000002
0.00000002
0.0000002
0.000002
0.00002
0.0002
0.002
0.02
0.2

anti-CD19 (BV421)

**Figure EV5. Variation in surface levels of CD19 on Nalm6 CombiCells produced by titration of Spytag-CD19.**

The indicated concentration of purified Spytag-CD19 was coupled to Nalm6 CombiCells before being detected by flow cytometry. Data information: A representative example out of 3 independent experiments is shown.

