## [Peer Review File · The EMBO Journal]

Using CombiCells, a platform for titration and combinatorial display of cell surface ligands, to study T-cell antigen sensitivity modulation by accessory receptors

Ashna Patel, Violaine Andre, Sofia Bustamante Eguiguren, Michael Barton, Jake Burton, Eleanor Denham, Johannes Pettmann, Alexander Morch, Mikhail Kutuzov, Jesús Siller-Farfan, Michael Dustin, Philip van der Merwe, and Omer Dushek

DOI: [10.15252/emboj.2023115263](https://doi.org/10.15252/emboj.2023115263)

Corresponding author: Omer Dushek (omer.dushek@path.ox.ac.uk)

Review Timeline:

Submission Date:	25th Aug 23
Editorial Decision:	28th Sep 23
Revision Received:	4th Oct 23
Editorial Decision:	1st Nov 23
Revision Received:	5th Nov 23
Accepted:	10th Nov 23

Editor: Kelly Anderson

Transaction Report:

Dear Prof. Dushek,

Thank you for submitting your manuscript for consideration by the EMBO Journal. It has now been seen by three referees whose comments are shown below.

Given the referees' positive recommendations, I would like to invite you to submit a revised version of the manuscript, addressing the comments of all three reviewers. I should add that it is EMBO Journal policy to allow only a single round of revision, and acceptance of your manuscript will therefore depend on the completeness of your responses in this revised version. I think it would be good to discuss your plan to address the referee concerns and I am available to do so by email or zoom in the coming weeks. I've also attached a guide for revisions for your convenience.

Thank you for the opportunity to consider your work for publication. I look forward to your revision.

Yours sincerely,

Kelly M Anderson, PhD
Editor, The EMBO Journal
k.anderson@embojournal.org

We realize that it is difficult to revise to a specific deadline. In the interest of protecting the conceptual advance provided by the work, we recommend a revision within 3 months (27th Dec 2023). Please discuss the revision progress ahead of this time with the editor if you require more time to complete the revisions.

Referee #1:

Measuring how T cell activation is regulated by varying the concentration and combination of cell surface ligands expressed on antigen-presenting cells (APC) remains labor intensive and subject to caveats linked to the fact that edited and/or transfected APC interaction may display some additional concealed variations in addition to the intended modification(s). The present report provides a novel platform enabling the rapid production of APC expressing any combination and concentration of ligands. The authors used it to compare T cell activation induced by native TCR, CARs, and bi-specific T cell engagers (BiTEs), and the contribution of accessory receptors such as CD28, LFA-1 and CD2.

The C-terminus of SpyCatcher - a protein of 12.3 kDa capable to form an intermolecular isopeptide bond with the 13 amino acids-long peptide SpyTag - was fused to the hinge segment of CD52 and expressed in CHO-K1 cells that were rendered negative for ICAM-1. CHO-K1 ICAM-1- hCD52-Spycatcher+ (CHO-K1 CombiCells) were then decorated with recombinant proteins corresponding to the full extracellular domains of CD58, ICAM-1, CD80, and CD86 fused to a C-terminal Spytag (for coupling to Spycatcher) and Histag (for purification). The absolute number of ligands that can be coupled exceeded ~ 106 per CHO-K1 CombiCells and they display normal mobility. Importantly, the surface level of Spytag-pMHC (the TCR ligand) can be varied without impacting the surface level of each Spytag-ligand (and vice versa). Moreover, in the case of combination of Spytag-proteins, a commensurate representation was reached at the cell surface provided that the total concentration of Spytag-ligands remained below 1 μ M. The authors then used this elegant model to compare T cell activation induced by native TCR, CARs, and BiTEs, and the contribution of accessory receptors such as CD28, LFA-1 and CD2. The authors published a recent paper addressing rather similar issues using a conventional (non SpyCatcher-based) approach (doi: 10.1073/pnas.2216352120). Importantly, they exploited their SpyCatcher-based approach to provide novel findings demonstrating (1) that Yescarta and Kymriah CAR-T cells can exploit LFA-1 but not CD2 or CD28 for improving their antigen sensitivity, (2) that the antigen

sensitivity of the TCR is higher than BiTEs, and (3) that PD-1 ligation directly inhibits TCR signalling and therefore, the ability of PD-1 to inhibit CD2 and CD28 co-stimulation may be a result of removing primary TCR signal. The power and results of this elegant methodological approach are convincing and further our understanding of T cell-based cancer immunotherapies, constituting an appropriate Resource paper.

Minor comments

1/ The manufacturing of the different T cell formats used by the authors needs a step of pre-activation. The authors need to specify it and considering that naive CAR-T cells have been recently developed, they may discuss whether their findings will also apply to naive CAR-T cells.

2/ The authors and other (Q. Xiao et al., Size-dependent activation of CAR-T cells. *Sci. Immunol.* 7, eabl3995 (2022)) have suggested that CARs triggering uses the kinetic segregation mechanism. Since Spycatcher has approximately the size of an Ig V or C domain does it mean that it enlarges the size of the engineered ligand-receptor pair by the length corresponding to an Ig V or C domain? Accordingly, will an 'enlarged' CD28-CD80 pair be capable of appropriately functioning in the context of 'normal' CD2-CD58 pairs?

3/ In the experiment comparing the activation induced by native TCR and CARs have the authors used cells with matched levels of TCR and CARs?

4/ I have difficulty to localize the Histag within the constructs?

Referee #2:

In this innovative and creative study, Patel et al adapt the Spycatcher-Spytag system to enable the coupling of combinations of proteins to the cell surface. By expressing the C-terminus of Spycatcher fused to the GPI-anchor domain of CD52 in cells, they are able to covalently link extracellular protein domains fused to a C-terminal Spytag to the cell surface. They demonstrate the potential for their 'CombiCells' system to provide insight into intercellular receptor-ligand interactions by fusing the extracellular domains of T-cell co-stimulatory ligands CD58, ICAM-1, CD80 and CD86 to their CombiCells and evaluating the effects on T-cell activation and cytokine secretion in a range of different T-cell co-culture systems (TCR, CARs and BiTEs). In addition, they evaluate the effects of PD-L1 CombiCell expression on engineered Jurkat T-cell activation.

This novel approach has potential to greatly facilitate mechanistic evaluation of receptor-ligand interactions. Key advantages are the ability to generate cells expressing multiple different ligands or ligand combinations and precisely titrate the level of ligand or antigen present at the cell surface. I believe that this manuscript will make an important contribution to the field, providing a platform for future work. As highlighted by the authors in the discussion, key limitations of the system are that ligands lack their native transmembrane and intracellular cytoplasmic domains, which is likely to affect their behaviour, membrane localisation and normal regulatory turnover. Nevertheless, the series of experiments presented in the paper demonstrates the potential of the system to reveal differential effects of different T cell co-stimulatory receptor/ligand interactions on T cell antigen sensitivity. The manuscript is well written and has a logical flow and the data are beautifully presented. The inclusion of diagrams in the figure panels to illustrate the experimental design is an extremely helpful addition. The experiments provide a comprehensive validation of the CombiCell system, include appropriate controls and are technically of a high standard. I only have a few minor comments and suggestions:

Figure S7 shows that 'Cytokine production by CD19-targeting CAR-T cells is largely independent of accessory receptors', however this is not currently referenced or discussed in the manuscript text i.e. in reference to figure 4 (p9). The text currently states that 'the antigen sensitivity of these CD19-targeting CARs was enhanced more by LFA-1 than by CD2 ligands (Fig. 4F).' However, this is based only on levels of surface 4-1BB and the authors should also discuss the cytokine production data in figure S7.

Figure 6A - This panel shows the effect of CombiCell expressed PD-L1 on co-cultured Jurkat cells. Il-8 levels are shown, whereas all preceding co-culture experiments used Il-2 levels (+/- IFNg and TNFa) as a measure of T-cell activation and cytokine production. Activated Jurkat cells also produce Il-2 so the authors should note, and explain the rationale for, the switch to using Il-8 as a readout for these experiments.

Referee #3:

Patel et al report the development of a cellular platform, termed "CombiCells", that would allow different combinations of cell surface ligands to be presented on a specifically engineered cell at titratable densities, so that the impact of these ligands on other cell types can be measured. This is done by engineering a cell line to express the Spycatcher protein on its surface, and then adding proteins labeled with the Spytag peptide, so that these will form a spontaneous covalent bond. As proof of concept the authors show the effect of different co-stimulatory receptors on T cells, particularly showing that TCR-transduced T cells and CAR T cells have differences in how they respond to CD58 and ICAM-1.

The authors have a clear rationale for this study and the experiments are logically presented. However, I have several concerns about the utility of this platform.

Major concerns

1. The authors state the primary benefit of CombiCells to be the ease and speed of adding different combinations of cell surface proteins at well-defined densities. However, each cell has a different native protein expression profile that needs to be considered - for this platform to work in the manner that they claim, one would theoretically need to start with a cell line that completely lacks cell surface protein expression, and so can be a blank canvas in which to add the different ligands. The authors do not have these cells, so they use CHO cells that are of hamster origin, but find that even these express ICAM-1 that can cross-react with human T cells, as shown in Fig S3. This required generation of ICAM-1 KO cells using CRISPR/Cas9, which is the very method that the authors compare and find themselves superior to. The authors then go on to use Nalm6 cells, a human B cell line, that required either CD19 or B2M KO to evaluate the activity of CAR T cells or TCR-T cells. As such, this is not a platform that can be deployed "within minutes" as the authors claim. Furthermore, besides the CRISPR-engineering required to remove native protein expression, there is an additional step of making the Spytag-labeled ligand proteins that needs to be considered - using the authors' methods as a reference, it seems that this would take a few weeks in itself, starting from making the transfer plasmid, transfecting cells, waiting for protein expression, then protein purification and verification. Therefore I would modify the claims to include these caveats for the work presented in this manuscript.

2. Figure 1H - it is unclear to me why the Spytag-labeled ligands degrade over time. If the ligands are covalently bound to a stable cell surface Spycatcher protein, why is the expression lost within 24 hours on the CHO cells? This may be a problem for investigating cell-cell interactions broadly, as this platform would only be relevant for short-term interactions.

3. Could the authors provide their rationale for looking specifically at ICAM-1, CD58, CD80 and CD86? There are many different T cell co-stimulatory ligands, and ICAM-1 and CD58 would not be the first to come to mind. More relevant might be 4-1BBL - differences in response to CD80/86 and 4-1BBL might be interesting when comparing CAR T cells with CD28 and 4-1BB co-stimulatory domains. Other relevant co-stimulatory receptors include CD70, CD40, OX40L.

4. Figure 3B - the authors state in the text that the TCR and CAR are expressed at similar levels, however the data shows that CAR expression is significantly higher.

5. Figure 4F - it would be of interest to see what the native expression of CD58 is on Nalm6 cells. If it is present, CD58 KO should decrease CAR T cell cytotoxicity. If it is absent, adding CD58 using the Spycatcher-Spytag system should increase cytotoxicity. This would be a more relevant readout than using CHO cells.

6. Figure 6 - why do the authors use Jurkat cells rather than primary human T cells for this one experiment? Also why is the expression of IL-8 and CD69 measured, when in all the prior figures (Fig 2-5) surface 4-1BB and IL-2 is reported?

7. Clinically, Yescarta and Kymriah have comparable patient outcomes which are superior to blinatumomab. However, in this paper blinatumomab was similar to Yescarta and superior to Kymriah in terms of T cell activation and cytotoxicity. This raises concern that the surrogate measures used in these studies may not reflect actual CAR T cell effectiveness when treating human subjects, and so the translational relevance of these assays is unclear.

Minor concerns

1. Methods lack sufficient detail. Source of the cell lines used are not identified. Lentivirus production methods are not specified. Also which cells were used for protein production is not clear.

2. Fig S6 - it would be interesting to see flow plots illustrating the CD19 expression levels on each Nalm6 cell clone, so that the reader can visualize the level of antigen the CAR T cells are recognizing.

Referee #1:

Measuring how T cell activation is regulated by varying the concentration and combination of cell surface ligands expressed on antigen-presenting cells (APC) remains labor intensive and subject to caveats linked to the fact that edited and/or transfected APC interaction may display some additional concealed variations in addition to the intended modification(s). The present report provides a novel platform enabling the rapid production of APC expressing any combination and concentration of ligands. The authors used it to compare T cell activation induced by native TCR, CARs, and bi-specific T cell engagers (BiTEs), and the contribution of accessory receptors such as CD28, LFA-1 and CD2. The C-terminus of SpyCatcher - a protein of 12.3 kDa capable to form an intermolecular isopeptide bond with the 13 amino acids-long peptide SpyTag - was fused to the hinge segment of CD52 and expressed in CHO-K1 cells that were rendered negative for ICAM-1. CHO-K1 ICAM-1⁻ hCD52-Spycatcher⁺ (CHO-K1 CombiCells) were then decorated with recombinant proteins corresponding to the full extracellular domains of CD58, ICAM-1, CD80, and CD86 fused to a C-terminal Spytag (for coupling to Spycatcher) and Histag (for purification). The absolute number of ligands that can be coupled exceeded ~ 10⁶ per CHO-K1 CombiCells and they display normal mobility. Importantly, the surface level of Spytag-pMHC (the TCR ligand) can be varied without impacting the surface level of each Spytag-ligand (and vice versa). Moreover, in the case of combination of Spytag-proteins, a commensurate representation was reached at the cell surface provided that the total concentration of Spytag-ligands remained below 1 μ M. The authors then used this elegant model to compare T cell activation induced by native TCR, CARs, and BiTEs, and the contribution of accessory receptors such as CD28, LFA-1 and CD2. The authors published a recent paper addressing rather similar issues using a conventional (non SpyCatcher-based) approach (doi: 10.1073/pnas.2216352120). Importantly, they exploited their SpyCatcher-based approach to provide novel findings demonstrating (1) that Yescarta and Kymriah CAR-T cells can exploit LFA-1 but not CD2 or CD28 for improving their antigen sensitivity, (2) that the antigen sensitivity of the TCR is higher than BiTEs, and (3) that PD-1 ligation directly inhibits TCR signalling and therefore, the ability of PD-1 to inhibit CD2 and CD28 co-stimulation may be a result of removing primary TCR signal. The power and results of this elegant methodological approach are convincing and further our understanding of T cell-based cancer immunotherapies, constituting an appropriate Resource paper.

We thank the reviewer for taking the time to read our manuscript in detail and providing constructive feedback.

Minor comments

1/ The manufacturing of the different T cell formats used by the authors needs a step of pre-activation. The authors need to specify it and considering that naive CAR-T cells have been recently developed, they may discuss whether their findings will also apply to naive CAR-T cells.

We have revised the results sections to specify the source of T cells when first introduced (both primary T cells and Jurkat T cells).

We have included a discussion paragraph on limitations to explain that different T cell subsets express different combinations of accessory receptors. We note that naïve T cells express CD2, LFA-1, and CD28, and therefore, it is reasonable to expect that our molecular findings using T cell blasts will translate to naïve T cell. Indeed, early work on CD2 and LFA-1 has shown that they can impact antigen sensitivity in a similar way on naïve T cells (Bachmann et al (1999) J. Exp. Med PMID: 10562314 & Bachmann et al (1997) Immunity PMID: 9354475). It follows that transducing CARs into naïve T cells before inducing their expansion is likely to result in CAR-T cells that are also inefficient at exploiting CD2 co-stimulation.

2/ The authors and other (Q. Xiao et al., Size-dependent activation of CAR-T cells. *Sci. Immunol.* 7, eabl3995 (2022)) have suggested that CARs triggering uses the kinetic segregation mechanism. Since SpyCatcher has approximately the size of an Ig V or C domain does it mean that it enlarges the size of the engineered ligand-receptor pair by the length corresponding to an Ig V or C domain? Accordingly, will an 'enlarged' CD28-CD80 pair be capable of appropriately functioning in the context of 'normal' CD2-CD58 pairs?

The distance between the c-terminus of the Spytag-ligand and the membrane when coupled to surface Spycatcher is difficult to determine. The schematic we are showing in Fig 1A is our most conservative estimate, which as the reviewer notes could add approximately one Ig domain. However, if surface Spycatcher adopts different orientations it will present the ligand closer to the membrane. Furthermore, while we used the full extracellular region in our Spytag ligand, it is possible to shorten the native ligand hinge, relying on the hinge functionality of the Spycatcher fusion proteins. Nonetheless and as the reviewer notes, an increase in size of ligands coupled to surface spycatcher is expected.

To avoid differences in size, we focused our comparisons of the impact of CD2, CD28, LFA-1, and PD-1 when their ligands were coupled to surface Spycatcher along with the appropriate antigen on CHO-K1 CombiCells. As a result, any increase in size would equally apply antigen and any presented ligands. Therefore, we do not have any conditions in the manuscript where CD28/CD80 are different in dimensions to CD2/CD58.

However, even if sizes are matched an overall increase in size may reduce CD45 segregation. In previous work from our lab, we have found that increasing antigen size by a minimum of 2 Ig domains dramatically reduced T cell activation (e.g. Choudhuri et al (2005) *Nature*, doi: 10.1038/nature03843) but in the present work, we have observed robust antigen recognition ($EC_{50} \sim 10^{-6}$ uM). This suggests that any impairment of CD45 segregation is more modest than in our previous work, which is likely a result of the compact dimension of the surface Spycatcher.

We have included a new discussion paragraph highlighting the kinetic-segregation model and the implications it has on presenting antigen on surface Spycatcher. We have also noted that the increase in size could be mitigated by appropriately shortening the endogenous hinge, taking advantage of the hinge functionality provided by the SpyCatcher-hinge fusion proteins.

3/ In the experiment comparing the activation induced by native TCR and CARs have the authors used cells with matched levels of TCR and CARs?

When using the pMHC-specific CAR, we were able to compare TCR and CAR expression using the same pMHC-tetramer finding slightly higher expression for the CAR (Fig 3B). This is consistent with our previous extensive analysis of a large number of different CARs finding higher surface expression compared to the TCR whose expression is likely limited by the requirement for other CD3-subunits (see Fig S2D in Burton et al (2013) PNAS). When using the clinical CD19-specific CARs (Yescarta and Kymriah), it is not possible to directly compare expression with the pMHC-specific TCR.

We have included this limitation in the 3rd discussion paragraph.

4/ I have difficulty to localize the Histag within the constructs?

The Histag is c-terminal to the Spytag.

We have revised the methods to explicitly state this.

Referee #2:

In this innovative and creative study, Patel et al adapt the Spycatcher-Spytag system to enable the coupling of combinations of proteins to the cell surface. By expressing the C-terminus of Spycatcher fused to the GPI-anchor domain of CD52 in cells, they are able to covalently link extracellular protein domains fused to a C-terminal Spytag to the cell surface. They demonstrate the potential for their 'CombiCells' system to provide insight into intercellular receptor-ligand interactions by fusing the extracellular domains of T-cell co-stimulatory ligands CD58, ICAM-1, CD80 and CD86 to their CombiCells and evaluating the effects on T-cell activation and cytokine secretion in a range of different T-cell co-culture systems (TCR, CARs and BiTEs). In addition, they evaluate the effects of PD-L1 CombiCell expression on engineered Jurkat T-cell activation. This novel approach has potential to greatly facilitate mechanistic evaluation of receptor-ligand interactions. Key advantages are the ability to generate cells expressing multiple different ligands or ligand combinations and precisely titrate the level of ligand or antigen present at the cell surface. I believe that this manuscript will make an important contribution to the field, providing a platform for future work. As highlighted by the authors in the discussion, key limitations of the system are that ligands lack their native transmembrane and intracellular cytoplasmic domains, which is likely to affect their behaviour, membrane localisation and normal regulatory turnover. Nevertheless, the series of experiments presented in the paper demonstrates the potential of the system to reveal differential effects of different T cell co-stimulatory receptor/ligand interactions on T cell antigen sensitivity.

The manuscript is well written and has a logical flow and the data are beautifully presented. The inclusion of diagrams in the figure panels to illustrate the experimental design is an extremely helpful addition. The experiments provide a comprehensive validation of the CombiCell system, include appropriate controls and are technically of a high standard.

We thank the reviewer for taking the time to read our manuscript in detail and providing constructive feedback.

I only have a few minor comments and suggestions:

Figure S7 shows that 'Cytokine production by CD19-targeting CAR-T cells is largely independent of accessory receptors', however this is not currently referenced or discussed in the manuscript text i.e. in reference to figure 4 (p9). The text currently states that 'the antigen sensitivity of these CD19-targeting CARs was enhanced more by LFA-1 than by CD2 ligands (Fig. 4F).' However, this is based only on levels of surface 4-1BB and the authors should also discuss the cytokine production data in figure S7.

We have included a new paragraph to discuss these conclusions.

Figure 6A - This panel shows the effect of CombiCell expressed PD-L1 on co-cultured Jurkat cells. Il-8 levels are shown, whereas all preceding co-culture experiments used Il-2 levels (+/- IFN γ and TNF α) as a measure of T-cell activation and cytokine production. Activated Jurkat cells also produce Il-2 so the authors should note, and explain the rationale for, the switch to using Il-8 as a readout for these experiments.

We are using a E6.1 Jurkat T cell clone where CRISPR has been used to knockout the TCRalpha/beta-chains. This clone produces low levels of IL-2, which is why we switched to measuring IL-8. We note that our conclusions using 4-1BB and IL-2 in the main text (e.g. Fig 2 & 3) are the same when examining CD69 and other cytokines (e.g. Fig S4, S5).

We have revised the results section to explain the rationale for using Jurkat T cells and the discussion to explain why we used IL-8/CD69.

Referee #3:

Patel et al report the development of a cellular platform, termed "CombiCells", that would allow different combinations of cell surface ligands to be presented on a specifically engineered cell at titratable densities, so that the impact of these ligands on other cell types can be measured. This is done by engineering a cell line to express the Spycatcher protein on its surface, and then adding proteins labeled with the Spycatcher peptide, so that these will form a spontaneous covalent bond. As proof of concept the authors show the effect of different co-stimulatory receptors on T cells, particularly showing that TCR-transduced T cells and CAR T cells have differences in how they respond to CD58 and ICAM-1.

The authors have a clear rationale for this study and the experiments are logically presented. However, I have several concerns about the utility of this platform.

We thank the reviewer for taking the time to read our manuscript in detail and providing constructive feedback.

Major concerns

1. The authors state the primary benefit of CombiCells to be the ease and speed of adding different combinations of cell surface proteins at well-defined densities. However, each cell has a different native protein expression profile that needs to be considered - for this platform to work in the manner that they claim, one would theoretically need to start with a cell line that completely lacks cell surface protein expression, and so can be a blank canvas in which to add the different ligands. The authors do not have these cells, so they use CHO cells that are of hamster origin, but find that even these express ICAM-1 that can cross-react with human T cells, as shown in Fig S3. This required generation of ICAM-1 KO cells using CRISPR/Cas9, which is the very method that the authors compare and find themselves superior to. The authors then go on to use Nalm6 cells, a human B cell line, that required either CD19 or B2M KO to evaluate the activity of CAR T cells or TCR-T cells. As such, this is not a platform that can be deployed "within minutes" as the authors claim. Furthermore, besides the CRISPR-engineering required to remove native protein expression, there is an additional step of making the Spycatcher-labeled ligand proteins that needs to be considered - using the authors' methods as a reference, it seems that this would take a few weeks in itself, starting from making the transfer plasmid, transfecting cells, waiting for protein expression, then protein purification and verification. Therefore I would modify the claims to include these caveats for the work presented in this manuscript.

We understand the reviewers concern and moreover, timescales can vary depending on the laboratory expertise, availability of reagents, etc. The 'minutes' applies to our lab because we have the CHO-K1 CombiCells and a library of purified Spycatcher-ligands that we're interested in studying (and are available to any scientist). But this may not be the case for other laboratories focused on other ligands.

A relevant timescale that is independent of the specific laboratory is the time that cells displaying different ligands are in culture before they are used for an experiment. The CombiCells are only in culture for 'minutes' after the independent conditions are produced

(i.e. after coupling different ligand combinations) and before they are used for experiments. This removes any issues of genetic drift that can happen in the standard approach, where large numbers of different cells lines are cultured separately and compared 'weeks' later.

We have revised the 2nd discussion paragraph to reflect more accurately that using CombiCells reduces the timescale that cells displaying different ligands are in culture from weeks to minutes, reducing genetic drift between cell lines. We removed "time to produce" from the graphical abstract. We left the sentence "Generate cells expressing different ligands at different concentrations within minutes" under the CombiCell platform schematic in the graphical abstract because once the cell and ligands are available, any combination/concentration of ligand presenting cell can be produced within minutes.

2. Figure 1H - it is unclear to me why the Spytag-labeled ligands degrade over time. If the ligands are covalently bound to a stable cell surface Spycatcher protein, why is the expression lost within 24 hours on the CHO cells? This may be a problem for investigating cell-cell interactions broadly, as this platform would only be relevant for short-term interactions.

The loss of Spytag-ligand on the cell surface is a result of the turnover of Surface Spycatcher. Although the total amount of surface Spycatcher is constant (Fig. S2B), individual molecules are routinely produced and degraded which means that the initial 'bolus' of Spytag-ligand that is coupled to the cell surface is lost over a day. This turnover is cell-type specific and for example, it is much slower on Nalm6 cells (Fig 4C).

We revised the paragraph on limitations to more clearly explain the implications of this turnover of surface Spycatcher.

3. Could the authors provide their rationale for looking specifically at ICAM-1, CD58, CD80 and CD86? There are many different T cell co-stimulatory ligands, and ICAM-1 and CD58 would not be the first to come to mind. More relevant might be 4-1BBL - differences in response to CD80/86 and 4-1BBL might be interesting when comparing CAR T cells with CD28 and 4-1BB co-stimulatory domains. Other relevant co-stimulatory receptors include CD70, CD40, OX40L.

Our study was focused on antigen sensitivity and therefore, we wanted to select ligands that are known to impact TCR sensitivity and determine their impact on CARs/BiTEs. We selected CD58 and ICAM-1 because in our previous work using plate-immobilised ligands, we found that they had the largest impact on antigen sensitivity for the TCR (Burton et al (2023) PNAS, [10.1073/pnas.2216352120](https://doi.org/10.1073/pnas.2216352120)). We previously examined co-stimulation by TNFRSF members but found that they did not impact on antigen sensitivity (Nguyen et al (2021) Mol Syst Biol, [10.15252/msb.202110560](https://doi.org/10.15252/msb.202110560)).

We have revised the discussion to highlight the reason for focusing on antigen sensitivity and CD2/LFA-1/CD28.

4. Figure 3B - the authors state in the text that the TCR and CAR are expressed at similar

levels, however the data shows that CAR expression is significantly higher.

We have revised the description to read “and using pMHC-tetramers, we found that it expresses at slightly higher levels compared to the TCR”. We initially used the word “similar” and not “identical” expression because there was a significant difference but with a modest effect size (i.e. although it is significantly different the actual increase in surface expression of the CAR is only 1.3-fold).

5. Figure 4F - it would be of interest to see what the native expression of CD58 is on Nalm6 cells. If it is present, CD58 KO should decrease CAR T cell cytotoxicity. If it is absent, adding CD58 using the Spycatcher-Spytag system should increase cytotoxicity. This would be a more relevant readout than using CHO cells.

These experiments are challenging because normal APCs contain many ligands that can have compensatory phenotypes. As a result, removing a single ligand from an APC may not have a phenotype. The Nalm6 is a B cell line that expresses all the ligands we have studied, including CD58, ICAM-1, CD80/CD86, and PD-L1. In previous work we blocked CD58 on Nalm6 using antibodies but found only modest impact on the TCR and CAR (see Fig S8 of Burton et al (2023) PNAS). Whereas on CHO-K1 CombiCells we found that CD58 can increase antigen sensitivity by 200-fold, blocking CD58 on Nalm6 had a very modest 2-fold reduction in sensitivity. As explained above, Nalm6 are expected to have many ligands to T cell surface receptors with compensatory phenotypes making it difficult to study the contribution of a single receptor. This difficulty in studying the contribution of individual ligands in normal APCs was part of the motivation in developing the CHO-K1 CombiCells, which do not express common ligands to T cell surface proteins.

6. Figure 6 - why do the authors use Jurkat cells rather than primary human T cells for this one experiment? Also why is the expression of IL-8 and CD69 measured, when in all the prior figures (Fig 2-5) surface 4-1BB and IL-2 is reported?

We have used Jurkat T cells because there is presently no robust assay for PD-1 function in primary human T cells and it is an established assay for studying the molecular function of PD-1, see for example:

<https://www.ncbi.nlm.nih.gov/pmc/articles/PMC7265324/>

<https://www.ncbi.nlm.nih.gov/pmc/articles/PMC6581740/>

<https://elifesciences.org/articles/74276>

We are using a E6.1 Jurkat T cell clone where CRISPR has been used to knockout the TCRalpha/beta-chains. This clone produces very low levels of IL-2, which is why we switched to measuring IL-8. We note that our conclusions using 4-1BB and IL-2 in the main text (e.g. Fig 2 & 3) are the same when examining CD69 and other cytokines (e.g. Fig S4,S5) when using primary T cells.

We have revised the results section to explain the rationale for using Jurkat T cells when initially introduced and to explain the readouts.

7. Clinically, Yescarta and Kymriah have comparable patient outcomes which are superior to

blinatumomab. However, in this paper blinatumomab was similar to Yescarta and superior to Kymriah in terms of T cell activation and cytotoxicity. This raises concern that the surrogate measures used in these studies may not reflect actual CAR T cell effectiveness when treating human subjects, and so the translational relevance of these assays is unclear.

Our study has focused on antigen sensitivity of these different therapeutic modalities whereas overall clinical success is dependent on many factors that we have not assessed (e.g. exhaustion, persistence, etc). We have not claimed that antigen sensitivity is the only factor but it is now understood to be important in certain clinical scenarios, such as relapse (e.g. Majnzer & Mackall (2018) Cancer Discovery, <https://doi.org/10.1158/2159-8290.CD-18-0442>). We have highlighted the importance of antigen sensitivity in scenarios where antigen levels are low, including in infection and CAR-T cell relapse, in the introduction.

Minor concerns

1. Methods lack sufficient detail. Source of the cell lines used are not identified. Lentivirus production methods are not specified. Also which cells were used for protein production is not clear.

We have revised the methods to include the source of the HEK293 cells for protein production.

The production of lentivirus can be found in the first paragraph of “Production of TCR and CAR transduced primary human CD8+ T cells”.

The source of the CHO-K1 cells (ATCC) and Nalm6 cells (kind gift from Crystal Mackall) can be found in the manuscript.

2. Fig S6 - it would be interesting to see flow plots illustrating the CD19 expression levels on each Nalm6 cell clone, so that the reader can visualize the level of antigen the CAR T cells are recognizing.

We have include the flow cytometry histograms in Fig EV4A. We also include the equivalent flow cytometry when coupling CD19 via surface Spycatcher (Fig EV5).

Dear Omer,

Congratulations on a great revision. You will see below that two of the referees are fully satisfied with the revision, while one referee remains unconvinced of the utility of your system. I therefore consulted further with the first two referees for their opinion specifically on the utility and provide those comments as well. As you will see below, those additional comments are quite positive and we are therefore happy to move forward with your manuscript. That said, there was a concern raised regarding the use of IL-8 that I ask you to (non-experimentally) address in a revised version. When you submit your revised version, please also take care of the following editorial items and add this also to your point-by-point response:

1. Please reduce the number of keywords to 5.
2. Please remove the author contribution section from the main manuscript.
3. Please review our new policy on conflict of interest on the EMBO author guide website and update the title of this section to: Disclosure and competing interests statement.
4. For references, only the first 10 author names should be displayed. In cases where there are more than 10 authors, add et al. after the 10th author name.
5. Thank you for providing a synopsis figure, but please adjust the format to 550 pixels wide by 200-440 pixels high.
6. We include a synopsis of the paper (see website for examples). Please provide me with a general summary statement and 3-5 bullet points that capture the key findings of the paper.
7. All figures must be referred to in the main manuscript and in chronological order. Please include a reference to all of the EV figures in the revised version.
8. Please remove the one-sentence summary and the two open access statements from the main manuscript.
9. Please move the acknowledgement and Disclosure statements after the materials and methods.
10. In the figure legends, please include N information for 1f, 1h, 3b, 3e, 4c, 6a-c, and EV3a.
11. For 4C, we do not allow statistics on N=2 data, please remove the error bars and ensure that individual points are displayed.
12. Please rename Figure Captions to Figure Legends.
13. Each figure legend should contain a title, a data information statement, and the format should be updated according to our author guidelines online.

Thank you for the opportunity to consider your work for publication. I look forward to your revision.

Kind regards,

Kelly

Kelly M Anderson, PhD
Editor, The EMBO Journal
k.anderson@embojournal.org

Further information is available in our Guide For Authors: <https://www.embopress.org/page/journal/14602075/>

authorguide

Referee #1:

The authors have very appropriately addressed the issues I raised as well as those of the two other reviewers. In my view the paper fits the standards expected for The EMBO Journal.

Referee #2:

The authors have addressed all my previous comments. I have no further concerns.

Referee #3:

I remain unimpressed by the utility of this technology, and whether it can generate insights that have biological relevance.

The authors present their CombiCell platform as a universal method to study cell-cell interactions mediated by cell surface ligands. However, the only cell type that they interrogate is T cells, and even here they resort to using an artificial Jurkat cell line to evaluate the function of PD-1. They state that the assay cannot be performed using primary human T cells because there is no established readout of PD-1 function in these. I expect that for most cell surface proteins there will be no established readout in which the CombiCell platform can be utilized effectively. Therefore, this may only be useful for looking at stimulation of artificial cell lines with artificial readouts, and whether this has any true biological significance is unclear.

I also continue to question the use of different cytokines to measure activation in different situations. The authors state that the E6.1 Jurkat cells do not produce robust IL-2, so they switched to measuring IL-8. Convenience of measurement is not an adequate reason to choose a cytokine measurement, there should be biological justification. IL-8 is not a cytokine that is commonly used to assess T cell function. Additionally, none of the other main or supplementary figures using human T cells contain any measurement of IL-8, so I am not sure whether primary T cells secrete this cytokine at all.

Additional Comments from Referee 1: "Referee 3 is too extreme in his analysis of the paper. It is a quite elegant approach that can speed up analysis of the mechanisms of T cell recognition and the authors spent a fair amount of time illustrating its usefulness for T cells. It sounds a bit unfair to ask them to apply it in the frame of a single paper to other cell types. Concerning the use of IL-8 as a readout for T cell activation, it is indeed an unusual way of gauging Jurkat T cell activation and whether it has a physiological relevance is not discussed. I am more concerned however by the fact that the maximum amounts of IL-8 produced is rather small as compared to the maximum amounts of IL-2 produced by 'regular' Jurkat T cells."

Additional Comments from Referee 2: "I understand the referee's concerns that the platform described in the manuscript is an artificial system. However, a strength is the capacity to evaluate effects of single ligands and combinations of ligands at varying concentrations, and the data appear reproducible and robust. I believe that this system would complement other assays e.g. using knockout of individual ligands, which are less artificial but have more limited flexibility e.g. in being able to titrate ligand dose and easily explore combinatorial effects. As the authors say, evaluating PD-1 function in primary T-cells in vitro is difficult and therefore they use an engineered Jurkat T-cell line. I agree with the referee that this is an artificial readout. However, they show robust effects of co-stimulatory molecules on primary T-cell activation in earlier experiments, so I don't think it follows that the system would be restricted to artificial readouts. In my view the paper is suitable for publication in The EMBO Journal."

Referee #1:

The authors have very appropriately addressed the issues I raised as well as those of the two other reviewers. In my view the paper fits the standards expected for The EMBO Journal.

We thank the reviewer for taking the time to read our revisions.

Referee #2:

The authors have addressed all my previous comments. I have no further concerns.

We thank the reviewer for taking the time to read our revisions.

Referee #3:

I remain unimpressed by the utility of this technology, and whether it can generate insights that have biological relevance.

The authors present their CombiCell platform as a universal method to study cell-cell interactions mediated by cell surface ligands. However, the only cell type that they interrogate is T cells, and even here they resort to using an artificial Jurkat cell line to evaluate the function of PD-1. They state that the assay cannot be performed using primary human T cells because there is no established readout of PD-1 function in these. I expect that for most cell surface proteins there will be no established readout in which the CombiCell platform can be utilized effectively. Therefore, this may only be useful for looking at stimulation of artificial cell lines with artificial readouts, and whether this has any true biological significance is unclear.

In the present work we have shown that CD2, LFA-1, and CD28 can be used in primary human T cells and we have shown two therapeutic modalities can be used using primary T cells (CARs and BiTEs). The lack of functional readout for PD-1 in primary T cells is not a limitation of CombiCells but a limitation of the primary T cells that we and other laboratories can produce in vitro. Therefore, it is unclear why the inability to study PD-1 in primary T cells limits the use of CombiCells. On the contrary, CombiCells has resolved a major debate on the ability of PD-1 to inhibit T cell activation by being able to present different ligand combinations.

I also continue to question the use of different cytokines to measure activation in different situations. The authors state that the E6.1 Jurkat cells do not produce robust IL-2, so they switched to measuring IL-8. Convenience of measurement is not an adequate reason to choose a cytokine measurement, there should be biological justification. IL-8 is not a cytokine that is commonly used to assess T cell function. Additionally, none of the other main or supplementary figures using human T cells contain any measurement of IL-8, so I am not sure whether primary T cells secrete this cytokine at all.

We have used the production of IL-8 from Jurkat T cells as a second readout of T cell activation to study the molecular mechanism of PD-1 function. The observation that IL-8 and CD69 produced the same results in the Jurkat T cells, and that all readouts we have used in primary T cells (CD69/4-11B surface markers and IFN γ /IL-2/TNF α cytokines) also phenocopied each other, provides strong evidence that IL-8 in Jurkat T cells is a reasonable proxy to study how surface receptors (TCR, CD2, CD28, PD-1) regulate T cell activation. If we

observed a different result between IL-8 and CD69 in Jurkat T cells, we agree that further investigation would be required but this was not the case. In summary, we have not made any claims about PD-1 regulating IL-8 specifically but rather on how PD-1 regulates CD2 and CD28.

We have revised the results to more clearly explain that we are using IL-8 as a measure of T cell activation.

Additional Comments from Referee 1: "Referee 3 is too extreme in his analysis of the paper. It is a quite elegant approach that can speed up analysis of the mechanisms of T cell recognition and the authors spent a fair amount of time illustrating its usefulness for T cells. It sounds a bit unfair to ask them to apply it in the frame of a single paper to other cell types. Concerning the use of IL-8 as a readout for T cell activation, it is indeed an unusual way of gauging Jurkat T cell activation and whether it has a physiological relevance is not discussed. I am more concerned however by the fact that the maximum amounts of IL-8 produced is rather small as compared to the maximum amounts of IL-2 produced by 'regular' Jurkat T cells."

We thank the reviewer for taking the time to read the comments from Reviewer 3.

Regarding physiological role of IL-8, we highlight that Jurkat T cells are a cancer line that have many differences to primary T cells. As a result, the community has largely focused on using Jurkat T cells to study surface receptor proximal events and have used the downstream readouts as measures of T cell activation without making any claims on whether or not the downstream readouts are physiological. Put differently, we have not made any claims about PD-1 regulating IL-8 specifically but rather on how PD-1 regulates CD2 and CD28. We have revised the results to more clearly explain that we are using IL-8 as a measure of T cell activation.

The absolute amount of cytokine produced depends on many factors, including the total number of cells used, the effector:target ratios, and the concentration/combinations of ligands present on the target cell. We suggest that the low level of IL-8 may be a result of CombiCells displaying a limited defined set of ligands whereas conventional APCs display a large number of diverse ligands, and some of these may be important for co-stimulation of cytokine production. Importantly, the signal-to-noise in our IL-8 readout is high and all conclusions made based on IL-8 are reproduced with CD69.

Additional Comments from Referee 2: "I understand the referee's concerns that the platform described in the manuscript is an artificial system. However, a strength is the capacity to evaluate effects of single ligands and combinations of ligands at varying concentrations, and the data appear reproducible and robust. I believe that this system would complement other assays e.g. using knockout of individual ligands, which are less artificial but have more limited flexibility e.g. in being able to titrate ligand dose and easily explore combinatorial effects. As the authors say, evaluating PD-1 function in primary T-cells in vitro is difficult and therefore they use an engineered Jurkat T-cell line. I agree with the referee that this is an artificial readout. However, they show robust effects of co-stimulatory molecules on primary T-cell activation in earlier experiments, so I don't think it follows that the system would be

restricted to artificial readouts. In my view the paper is suitable for publication in The EMBO Journal."

We thank the reviewer for taking the time to read the comments from Reviewer 3.

Dear Omer,

Congratulations on an excellent manuscript, I am pleased to inform you that your manuscript has been accepted for publication in The EMBO Journal. Thank you for your comprehensive response to the referee concerns and for providing detailed source data. It has been a pleasure to work with you to get this to the acceptance stage.

I will begin the final checks on your manuscript before submitting to the publisher next week. Once at the publisher, it will take about 3 weeks for your manuscript to be available online. As a reminder, the entire review process, including referee concerns and your point-by-point response, will be available to readers.

I will be in touch throughout the final editorial process until publication. In the meantime, I hope you find time to celebrate!

Warm wishes,
Kelly

Kelly M Anderson, PhD
Editor, The EMBO Journal
k.anderson@embojournal.org
